

# Land-use and climate change effects on water yield from East African Forested Water Towers

Charles Nduhiu Wamucii[1], Pieter R. van Oel[2], Arend Ligtenberg[3], John Mwangi Gathenya[4], Adriaan J. Teuling[1]

[1]Hydrology and Quantitative Water Management Group, Wageningen University & Research, 6700 AA Wageningen, The Netherlands

[2]Water Resources Management Group, Wageningen University & Research, 6700AA Wageningen, The Netherlands

[3]Laboratory of Geo-information Science and Remote Sensing, Environmental Sciences, Wageningen University & Research, 6708 PB Wageningen, The Netherlands

[4]Soil, Water and Environmental Engineering Department, School of Biosystems and Environmental Engineering, Jomo Kenyatta
University of Agriculture & Technology, P.O Box 62000 - 00200 Nairobi, Kenya

*Correspondence to:* Charles Nduhiu Wamucii (charles.wamucii@wur.nl)

**Abstract.** East-African forested mountain regions are vital in generating and supplying water resources to adjacent
arid and semi-arid lowlands. However, these ecosystems are under pressure from both climate and land-use changes. This study aimed to analyze the effects of climate and land-use changes on water yield using the Budyko conceptual framework. For 9 selected forested water towers in East Africa, the amount and distribution of water resources and their decadal changes were analyzed. Results show that most areas inside and outside the water towers are under pressure from human influences. Water yield was observed to be more sensitive to climate
changes compared to land-use changes within the selected East African water towers themselves. However, for the surrounding lowlands, the effects of land-use changes have greater impacts on water yield. We conclude that the East-African water towers have seen a strong shift towards wetter conditions, especially in the period of 2011-2019 while at the same time, the atmospheric demand is gradually increasing. Given that majority of the water towers were identified as non-resilient to these changes, future water yield is likely to also experience more
extreme variations.

**Keywords:** Water towers, Climate changes, Land-use changes, Water yield, Budyko framework

## 1. Introduction

Many mountainous areas act as water towers by generating and supplying water resources to adjacent lowlands
that would otherwise be much drier. An area is considered a water tower if it has a high elevation and high precipitation, consequently generating streamflow to lowland areas (Dewi et al., 2017; Immerzeel et al., 2010; Viviroli et al., 2007). Although research on water towers has focused mainly on glaciated mountain chains (Immerzeel et al., 2020), there is growing awareness that forested mountains can provide similar services (Viviroli and Weingartner, 2004). Mountainous areas in Africa cover approximately 20 % of Africa's surface area and
receive significantly more rain than adjacent lowlands (EAC et al., 2016; UNEP, 2014). They capture water, store it, purify it, and release it to lowland areas (UNEP, 2014). The East-African region is one of the most mountainous areas of Africa with several peaks above 4,500 meters and hosts the three highest mountains on the continent: Kilimanjaro (5,895 m), Mount Kenya (5,119 m), and the Rwenzori Mountains (5,109 m) (UNEP, 2014).

Montane forest ecosystems in the East-African region are classified as water towers due to their high elevations
and high humidity thus generating water yield for adjacent lowland areas.  There is a high dependency on surface water in the East-African region (Jacobs et al., 2018), but rainfall distribution is meager in most parts of the region, with several areas experiencing frequent occurrence of severe droughts (Nicholson, 2017). Therefore, the forested water towers in the region are important sources of water that sustain environmental and human water demands in the lowland areas.

The water towers of East Africa are under pressure from human intensification and climate change (Gebrehiwot et al., 2014; WWF, 2005). According to the Intergovernmental Panel for Climate Change (IPCC) Fifth Assessment Report, the average annual temperature for Africa has risen by at least 0.5 °C during the last 100 years and this is predicted to increase by approximately 3.2 °C by 2080. This will dramatically diminish glaciers in East-African



water towers whose surface area has already decreased by 80 % since the 1990s (EAC et al., 2016), affecting runoff and water resources downstream. The East-African montane forest zones continue to be lost to agriculture and other anthropogenic uses. This is mainly attributed to high and increasing population density which is a major driving force of environmental change in the mountainous areas (UNEP, 2014).

Understanding historical climate change and human-induced land-use changes and their impacts on streamflow generation from the forested water towers can explain some of the extreme hydrological events experienced in the adjacent lowlands such as floods and hydrological droughts. To the best of our knowledge, there are no studies that have focused on the East-African forested water towers and their ability to generate streamflow under a changing climate and land-use in the East-African region. That said, Guzha et al. (2018) in their review emphasized

the importance of forests in streamflow generation in the region, with forest degradation leading to increased stream discharges and surface runoff. Moreover, Muthoni et al. (2019), focused on spatial-temporal trends and variability of precipitation within East and Southern Africa. However, there is limited information on the partitioning of the available precipitation into water yield and evapotranspiration from the forested water towers of the East-African region.

Various approaches have been used for studying the effects of climate change and land-use on streamflow. Jiang et al. (2015) categorized such methods into two; (a) deterministic rainfall-runoff models and (b) statistical methods. Dey and Mishra (2017) categorized these approaches into four categories; (i) experimental approaches, (ii) hydrological modeling, (iii) conceptual approaches, and (iv) analytical approaches. The Budyko model (Budyko, 1974) is a conceptual approach that considers both water and energy constraints in hydrological processes over a

long-term period. The framework has been applied to quantify or separate the impacts of climate change and human activities on runoff (Jiang et al., 2015; Roderick and Farquhar, 2011; Xu et al., 2013).

The Budyko framework has been applied successfully in numerous studies focusing on the partitioning of precipitation into streamflow and evapotranspiration (Creed et al., 2014; Jiang et al., 2015; Mwangi et al., 2016; Roderick and Farquhar, 2011; Xu et al., 2013; Zhang et al., 2004). The framework works well both at coarse global

grid resolution and in smaller basins of less than 10 km² (Redhead et al., 2016; Teuling et al., 2019; Zhang et al., 2004). This paper aims to analyze the effects of climate and land-use changes over the past decades and their impacts on the amount and distribution of water resources from selected forested water towers of East Africa. The water yield simulations were evaluated against observation-based runoff.

## 2.  Data and Methodology

The Budyko conceptual framework was adopted to evaluate the impacts of land-use changes and climate changes on water yield from the selected forested water towers. The study area is the East-African region. The montane forest ecosystems are the major

forest types in Eastern Africa. They range from Ethiopian highlands to Albertine rift mountains stretching along the Congo DRC and bordering Uganda, Rwanda, Burundi, and Tanzania. This study focused on montane forest ecosystems and their

moorlands. The selected water towers are shown in Fig 1, and summarized in Table 1).

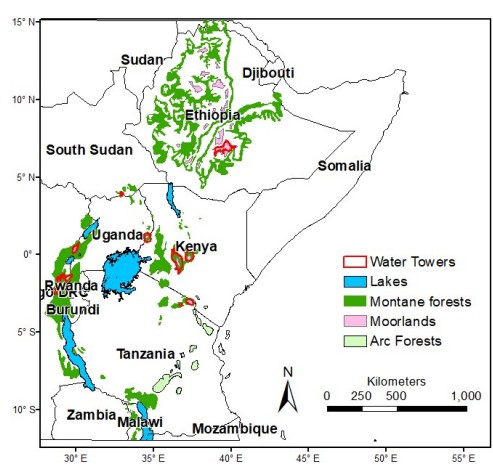

**Figure 1. The East African Forest Ecosystems and the location of the selected water towers**



**Table 1. The selected water towers of East Africa**

| Mountain Ecosystems | Location | Peak Elevation – asl (m) | Foot slope contour - asl (m) |
|---|---|---|---|
| Mt Kilimanjaro | Tanzania | 5,895 | 2000 |
| Mt Kenya | Kenya | 5,199 | 2000 |
| Mt Elgon | Kenya/Uganda | 4,321 | 2000 |
| Aberdare Ranges | Kenya | 3,999 | 2100 |
| Rwenzori Mountains | Uganda/Congo | 5,109 | 2000 |
| Mt Meru | Tanzania | 4,565 | 2000 |
| Virunga Mountains | Congo/Rwanda/Uganda | 4,507 | 2000 |
| Bale Mountains | Ethiopia | 4,337 | 2600 |
| Imatong Mountains | South Sudan/Uganda | 3,187 | 2000 |

Precipitation data (P) was gathered from Climate Hazards Group Infrared Precipitation with Stations (CHIRPS-v2) with a temporal coverage beginning 1981 and a spatial resolution of 0.05°. Potential Evapotranspiration (PET) data was sourced from Climate Research Unit (CRU) database with temporal coverage beginning 1981 and a spatial resolution of 0.5°. Normalized Difference Vegetation Index (NDVI) data to estimate land surface characteristics was sourced from Global Inventory Monitoring and Modeling System (GIMMS) Third Generation (3 g) Advanced Very High-Resolution Radiometer (AVHRR) sensor onboard the National Oceanic and Atmospheric Administration (NOAA) satellites at a spatial resolution of 0.07° (Kalisa et al., 2019). The research borrows from the concept of quantifying the long-term impact of climate and land-use changes on mean annual evapotranspiration and water yield at catchment scales based on data and parameters that are easily measurable at a regional scale (Zhang et al., 2001). Forested catchments generally have higher evapotranspiration than other land covers such as grassed catchments. Therefore changes in land use and forest management have an impact on catchment water balance and hence water yield (Teuling and Hoek van Dijke, 2020; Zhang et al., 2001).

One way of estimating water yield (Q) and actual evaporation (ET) is to assume that evapotranspiration from land surfaces is controlled by water availability and atmospheric demand (Zhang et al., 2001). The water availability can be approximated by precipitation; the atmospheric demand represents the maximum possible evapotranspiration and is often considered as the potential evapotranspiration (PET). Under very dry conditions, PET exceeds precipitation (P) and actual evapotranspiration (ETa) equals precipitation. Under very wet conditions water availability exceeds PET, and ET will asymptotically approach the potential evapotranspiration (Zhang et al., 2001), (see Fig A1 for key assumptions on energy and water limits). The Budyko Curve provides a "business as usual" reference condition for the water balance. Assuming that it can depict the expected partitioning of P into ET and Q, then it is possible to account for the reasons why some points depart from the baseline (Creed and Spargo, 2012b). The vertical deviations reflect a change in the partitioning of P into ET and Q, hence, the higher the evaporative index (EI), the less the streamflow (Q). The horizontal deviations reflect the change in climatic conditions (i.e. temperature and precipitation), thus, the higher the dryness index (DI), the warmer/drier the conditions. One important feature of the Budyko curve is the assumption that, under stationary conditions, hydrologic partitioning of study areas will follow on the Budyko Curve. However, under non-stationary conditions, each catchment will deviate from the Budyko curve depending on land cover and physical catchment characteristics (Creed and Spargo, 2012b), and this feature might be used to separate land cover change effects from climate change.

Several analytical equations have been proposed for the Budyko curve. In this study, FU's equation was used (Equation 1). The equation has been applied in different studies (Li et al., 2013; Teuling et al., 2019).

$$\frac{ET}{P} = 1 + \frac{PET}{P} - \left[1 + \left(\frac{PET}{P}\right)^{\omega}\right]^{1/\omega} \tag{1}$$

where *P*, *PET*, and *ET* are the precipitation, potential evaporation, and actual evapotranspiration. *PET/P* and *ET/P* are termed "dryness index" and "evaporation ratio", respectively.

Parameter ($\omega$) is an empirical parameter that controls how much of the available water will be evaporated given the available energy. It reflects the impact of other factors such as land surface characteristics and climate seasonality on water and energy balances (Li et al., 2013). Land surface hydrology varies due to variations in different factors such as vegetation, soil types, and topography, climate seasonality, etc (Li et al., 2013). Vegetation


information can serve as a good integrated indicator of these ecohydrological impacts on water and energy balances as it reflects the integrated landscape and climatic features (Donohue et al., 2007). The fact that vegetation coverage integrates the effects of the eco-hydrological processes on water and energy balances warrants a simple parameterization for ($\omega$) using only vegetation information in the large-scale basins (Li et al., 2013). According

to Li et al. (2013), ($\omega$) parameters can be calculated using the following equation:

$$\omega = 2.36M + 1.16 \tag{2}$$

Where ($M$) represents the vegetation coverage – which is calculated based on NDVI indices (Yang et al., 2009):

$$M = \frac{NDVI - NDVI_{min}}{NDVI_{max} - NDVI_{min}} \tag{3}$$

In this study, the M values were calculated for 1985, 1995, 2005, and 2015, and assumed to represent the ($\omega$) parameters for the respective periods of 1981-1990, 1991-2000, 2001-2010, and 201-2019.

Over a long period (i.e. 5-10 years) it is reasonable to assume that changes in soil water storage are zero (Creed et

al., 2014; Teuling et al., 2019; Zhang et al., 2001). Therefore, water yield is estimated using the following equation:

$$Q = P - ET \tag{4}$$

Where *(Q)* represents the water yield, *P* (precipitation), and *(ET)* the simulated actual evapotranspiration.

To develop the Budyko curves that are representative of the selected forested water towers, 100 random points were generated in each of the water towers in ArcGIS. The random points were used to extract values from raster P, PET, and ET grids into excel format for external analysis. For maximum representation, the minimum allowed

distance between the random points was set to 100 meters. The random points generated were assigned the respective values of PET, ET, and P using the Extract Multi Values to Points tool in ArcGIS. The Evaporative index (EI) values; calculated as a ratio of ETa and P, and Dryness index (DI) values; a ratio of PET and P were used to draw the Budyko curves. In this study, the Budyko curve for the 1981-1990 period was used as the reference condition for the water balance, to effectively assess the trends in the succeeding periods of 1991-1990,

1991-2000, 2001-2010, and 2011-2019.

To evaluate the impacts of climate and land-use changes, the sensitivity of climate and land-use changes was conducted. The climate and land-use values for the years 1981-1990 were used as the reference conditions in the Budyko framework. The climatic conditions (i.e. P and PET) for the years 1981-1990 were held constant in the Budyko framework to evaluate the impacts under changing land-use conditions in the succeeding periods of 1991-

2000, 2001-2010, and 2011-2019. Similarly, the land-use conditions (i.e. ($\omega$) parameters) for the years 1981-1990 were held constant in the Budyko framework to evaluate the impacts under changing climatic conditions in the succeeding periods of 1991-2000, 2001-2010, and 2011-2019.

Additionally, the deviations from the Budyko curves were also investigated to give a further understanding of the type of changes observed in the different water towers. Vertical deviations from the Budyko curve indicate

anthropogenic effects which result in increases or decreases in water yield (Creed and Spargo, 2012b, 2012a). The horizontal deviations reflect a shift to warmer or humid conditions mainly due to resultant variations in temperature and precipitation (Creed and Spargo, 2012b, 2012a). The deviation (d) and elasticity (e) are the two indices used to describe the potential departure from the theoretical Budyko curve of a catchment's DI and EI points with time (Creed et al., 2014). The deviation (d) was calculated using the following formula;

$$d = EI_{Sim} - EI_{Bud} \tag{5}$$

Where $EI_{Sim}$ represents EI simulated for periods in 1991-2000, 2001-2010 and 2011-2019 and $EI_{Bud}$ represents the predicted theoretical Budyko value for the reference period of 1981-1990. A negative *(d)* represents a downward shift from the Budyko curve and hence an increase in Q. A positive (d) represents an upward shift from the Budyko curve and hence a decrease in Q. The elasticity (e) was calculated as a ratio of DI ranges to EI ranges as shown in the following formula below;

$$e = \frac{\Delta DI}{\Delta EI} \tag{6}$$





Where $\Delta DI$ represents a range in DI values and $\Delta EI$ represents a range in EI values observed in the periods of 1991-2000, 2001-2010, and 2011-2019 using the period of 1981-1990 as the reference period. The water towers with lower elasticity values indicate greater ranges in their EI values and the water towers with higher elasticity values demonstrate lesser ranges in their EI values.

5    The simulated streamflow of the water towers was compared with composite runoff data downloaded from the Global Runoff Data Centre (GRDC). The composite runoff fields, developed through combining observed river discharge information with a climate-driven water balance model, provide the "best estimate" of terrestrial runoff over large domains (Fekete et al., 2002). A total of 312 points above 2000 meters above sea level, which is the focus of this study (i.e. elevated water towers), were randomly generated in ArcGIS. For maximum representation, 10   the minimum allowed distance between the random points was set to 100 meters. The selected random points and their respective values of simulated streamflow and composite runoff were compared.

## 3. Results

Higher long-term mean annual rainfall of above 1000 mm yr$^{-1}$ was observed majorly in the mountainous forest ecosystems located in the western region and the Ethiopian highlands in the north of East Africa (Fig 2a). The 15   mountainous forest ecosystems are important rainfall regions in drier environments as represented by Mt Kilimanjaro (average 1800 mm yr$^{-1}$), Mt Meru (average 1200 mm yr$^{-1}$), Mt Kenya (average 1400 mm yr$^{-1}$), and Aberdare ranges (average 1200 mm yr$^{-1}$) as shown in Fig 2a and Fig 2b. The 10-year moving averages analysis revealed patterns of high and low trends in precipitation in the different water towers (Fig A2).

Using the 1981-1990 period as the reference period, changes in Precipitation showed a Longitudinal gradient. 20   Negative changes in rainfall were observed in the water towers located towards the eastern side except for the Virunga mountains. Positive trends were observed in the water towers located towards the western side with exception of Aberdare ranges. Mt Kilimanjaro experienced a strong mean annual rainfall reduction with an average annual reduction of 13.5 % and 12 % observed in the periods of 2001-2010 and 2011-2019 respectively (Fig 2c). Contrastingly, a steady increase in mean annual rainfall was observed in Mt Elgon with an average increase of 25   over 20 % recorded in the years of 2011-2019 (Fig 2c).

Long-term assessment of atmospheric demand (PET) showed areas with relatively higher mean annual PET to coincide with areas of low rainfall and vice-versa (Fig 2d). Generally, atmospheric demand continued to increase over time in all the water towers with a peak observed in the years of 2001-2010 (Fig 2e and Fig A3). Imatong Mountains water tower had the highest atmospheric demand with an average long-term mean of appx 1500 mm 30   yr$^{-1}$, followed closely by Mt Elgon with an average long-term mean of appx 1400 mm yr$^{-1}$. The water towers located towards the western side exhibited lower atmospheric demand (examples are Virunga mountains – long-term mean of 990 mm yr$^{-1}$ and Rwenzori mountains – long-term mean of 1100 mm yr$^{-1}$).

Using the 1981-1990 period as the reference period, all water towers experienced increases in the annual atmospheric demand (Fig 2f.) Bale mountains saw a sharp increase in atmospheric demand by approximately 6 % 35   in the periods of 2001-2010 and 2011-2019. A minimal increase in atmospheric demand was observed in Mt Kilimanjaro with an average annual increase of 0.1 %, 1.0 %, and 0.8 % in the periods of 1991-2000, 2001-2010, and 2011-2019 respectively as shown in Fig 2f.

**Figure 2. Long-term Mean annual rainfall (a, b, c) and Potential Evapotranspiration (d, e, f) for the period 1981-2019**




Higher values for the Budyko parameter ($\omega$) were observed in the western part of Congo/Uganda, the Ethiopian Highlands and along the coastline of Tanzania, Kenya, and Somalia (Fig 3a). The land surface characteristics ($\omega$), ranged between 2.4 and 3.1 in the different water towers, with exception of Bale mountains where a drop to 2.3 was observed in 2015 (Fig 3b). Using the year 1985 as the reference for the land cover characteristics, different

patterns of negative changes and positive changes were observed. A major drop was observed in Mt Meru and Mt Kilimanjaro while Mt Elgon and Imatong mountains maintained a positive change (Fig 3c).

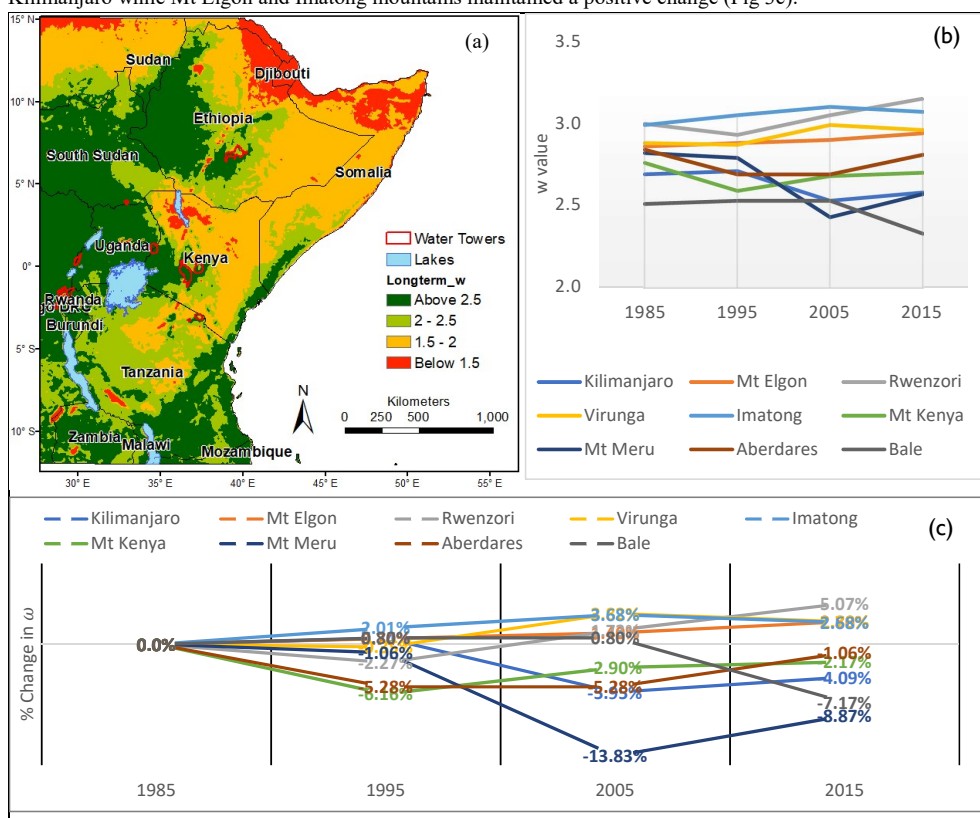

**Figure 3. The Land surface characteristics (ω) for 1985 to 2015 (a), the changes observed between 1985 and 2015 (b) & and (c)**

The long-term actual evapotranspiration (Eta) assessment revealed longitudinal differences in the spatial
distribution. The water towers towards the west were observed to be located in regions with higher ETa (examples are Mt Elgon, Imatong mountains, Rwenzori mountains, and Virunga mountains). The water towers towards the east are located in regions with relatively lower ETa (examples are Mt Meru, Mt Kilimanjaro, Aberdare Ranges, and Mt Kenya, (Fig 4a and Fig 5a, 5b, 5c, 5d). The changes in ETa in the region were analyzed using the 1981-1990 period as the baseline (Fig 4a). The decreases in ETa were observed in the South-Eastern parts of Ethiopia,
North-south gradient in Kenya, central Tanzania, and the western side of Congo/Burundi region as shown in Fig 4b, 4c, and 4d. The increases in ETa were observed in northern parts of Eastern Africa (i.e. Sudan, South Sudan, Djibouti, Northern Somalia, the Kenyan-Somali border, and parts of north-western Kenya bordering Uganda and South Sudan (Fig 4b, 4c, and 4d).



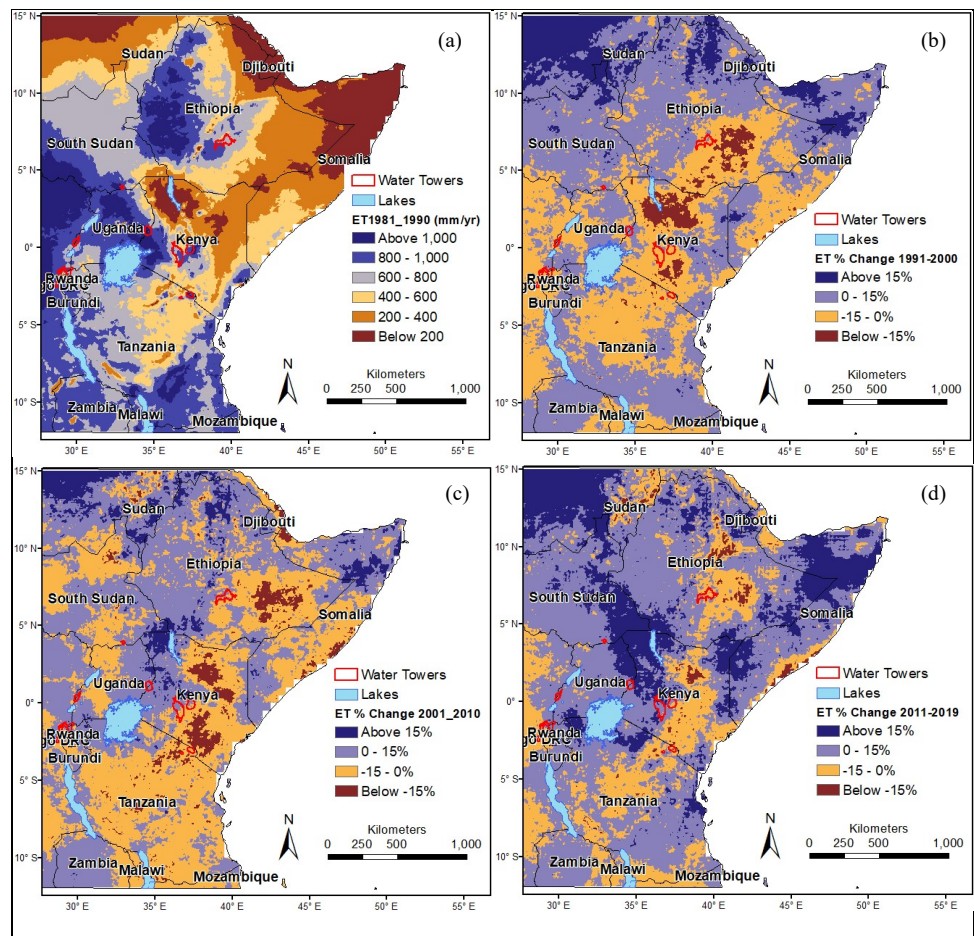

**Figure 4. Simulation of Actual Evapotranspiration (Eta) in the East African region:** (a) ETa for 1981-1990, (b) ETa changes in 1991-2000, (c) ETa changes in 2001-2010, and (d) ETa changes in 2011-2019

Despite the longitudinal differences, the individual water towers recorded varied ETa values. Higher ETa values were simulated around the Imatong Mountains – with a long-term mean of 1107 mm yr$^{-1}$, Mt Elgon – with long-term a mean of 1097 mm yr$^{-1}$, and Mt Kilimanjaro – with a long-term a mean of 1012 mm yr$^{-1}$. The lowest ETa values were observed in the Bale mountains – with a long-term mean of 747 mm yr$^{-1}$ (Fig 5j). Using the 1981-1990 period as the reference period, Mt Elgon recorded a steady increase in annual mean ET with an average increase of the order of 10 % observed between 2011 and 2019. Pronounced decreases in ET were observed in Mt Kilimanjaro and Mt Meru water towers (Fig 5k), consistent with the decreasing trend in precipitation.



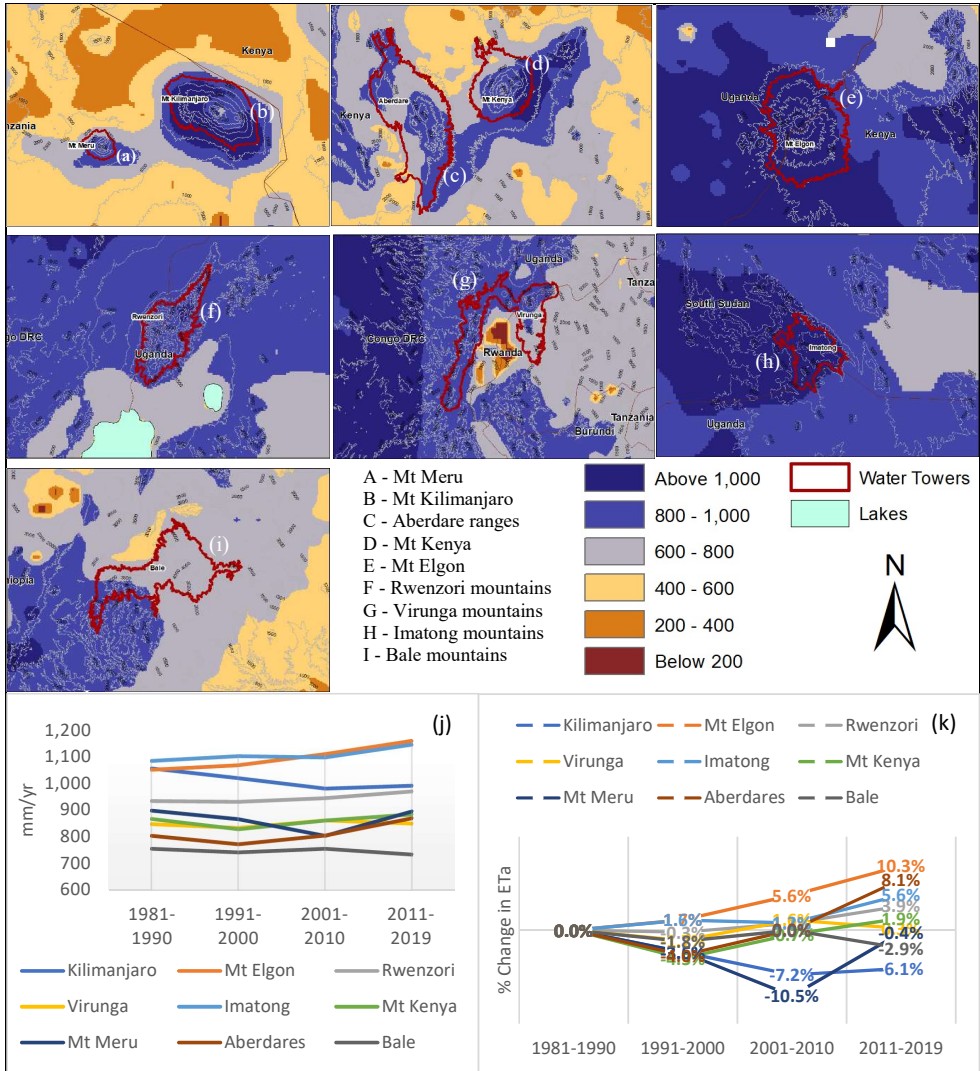

**Figure 5. The Long-term Actual Evapotranspiration (ETa) in and around the water towers (a-i) and Simulated Annual Mean Evapotranspiration in the water towers (j&k)**

Similarly to what was observed in ETa spatial distribution, the long-term water yield (Q) assessment also showed that the water towers located towards the east are surrounded by regions with low water yield potential below 200 mm yr-1 (examples are Mt Meru, Mt Kilimanjaro, Aberdare Ranges, and Mt Kenya (Fig 6a and Fig 7a, 7b, 7c, 7d). Using the period 1981-1991 as the baseline, major increases in Q were observed in areas of Sudan, and the Kenya-Somali border as shown in Fig 6b, 6c, and 6d.



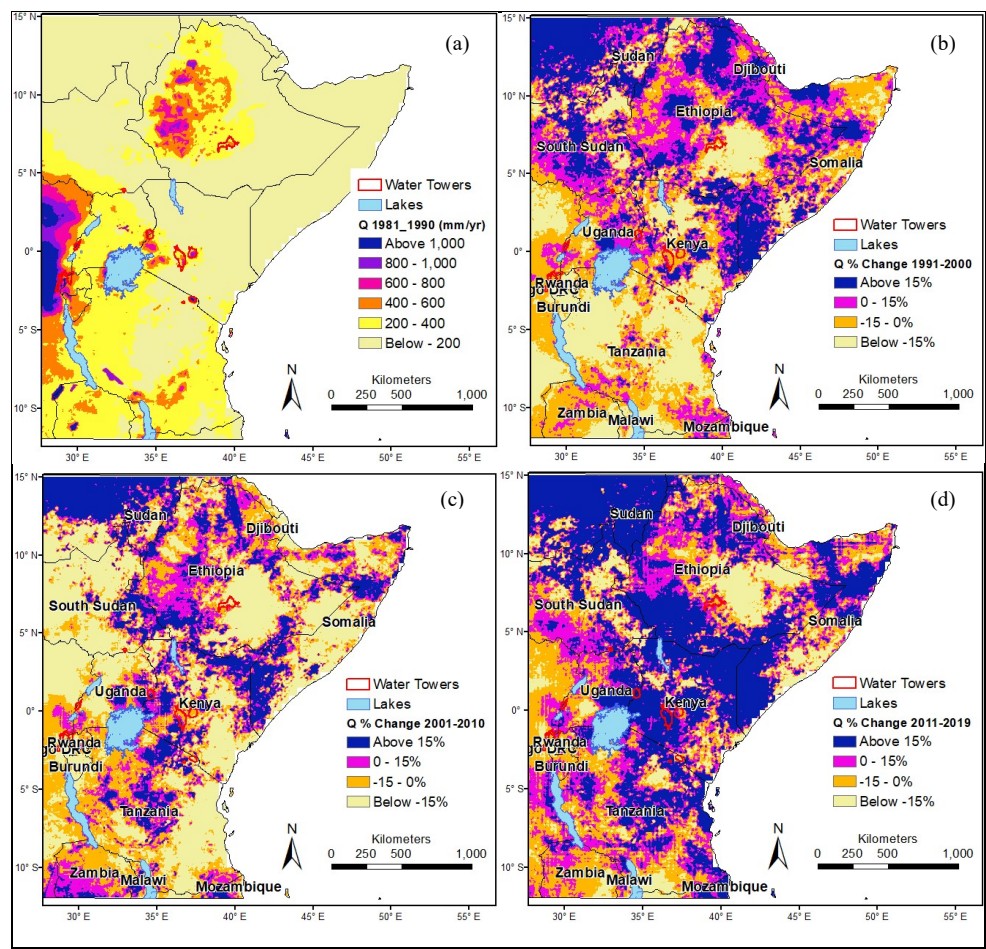

**Figure 6. Simulation of Water Yield (Q) in the East African region:** (a) **Q for 1981-1990,** (b) **Q changes in 1991-2000,** (c) **Q changes in 2001-2010, and** (d) **Q changes in 2011-2019**

Despite longitudinal differences, the higher mean annual water yield was however observed in Mt Kilimanjaro water tower located on the eastern side – with the long-term annual mean of 794 mm yr$^{-1}$, followed by two water
towers located on the western side (i.e. Virunga mountains – with a long-term annual mean of 676 mm yr$^{-1}$ and Rwenzori mountains – with a long-term annual mean of 650 mm yr$^{-1}$. The lowest annual mean water yield was observed in Bale mountains – with a long-term annual mean of 315 mm yr$^{-1}$ (Fig 7j). Conversion of water yield units from mm/yr to m$^3$/s per unit area revealed that Mt Kilimanjaro and Mt Kenya are important sources of water in the drier part of the East African region (Table A1).

Using the period 1981-1990 as the reference point, the positive and negative changes were observed in the different water towers. There was a consistent increase in annual mean water yield in Mt Elgon water tower with an order of 11.4 % and 42.9 % recorded in the periods of 2001-2010 and 2011-2019 respectively (Fig 7k). There was a decrease in water yield in the Aberdare and Mt Meru water towers during the 1991-2000 period, after which an increase in annual mean water yield was recorded in the later years. A consistent decline was observed in Mt
Kilimanjaro and the Virunga mountains during the study period.



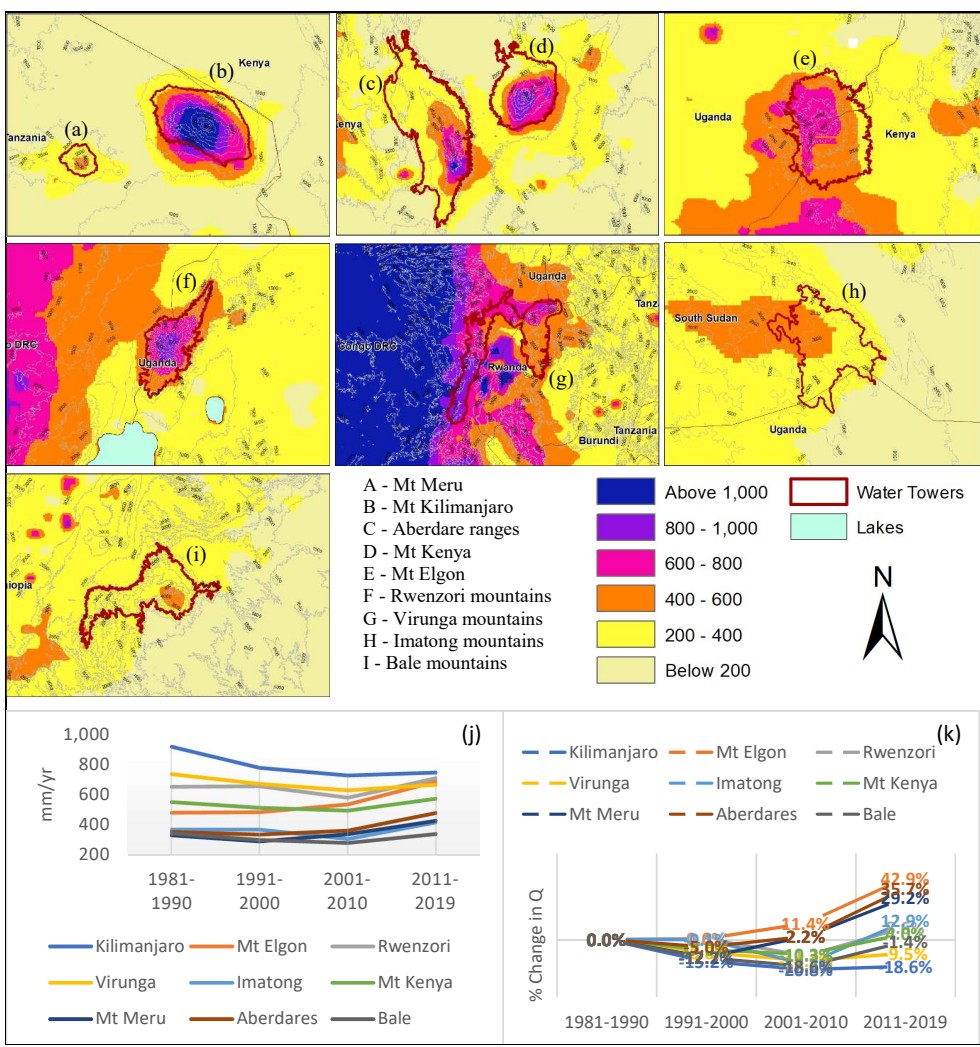

**Figure 7. The Long-term Water yield (Q) in and around the water towers (a-i) and Simulated Annual Mean Water Yield in the Water Towers (j&k)**

The simulated streamflow was compared with composite runoff data downloaded from the Global Runoff Data Centre (GRDC) (Fekete et al., 2002). The spatial pattern of the simulated streamflow closely resembles the pattern produced by GRDC composite runoff as shown in Fig 8a and 8b. A total of 312 points above 2000 meters above sea level, the focus of this study (i.e. elevated water towers), were randomly selected and their respective values of simulated streamflow and composite runoff were compared. The Kling-Gupta efficiency test revealed positive values, KGE=0.33 as shown in (Fig 8c).



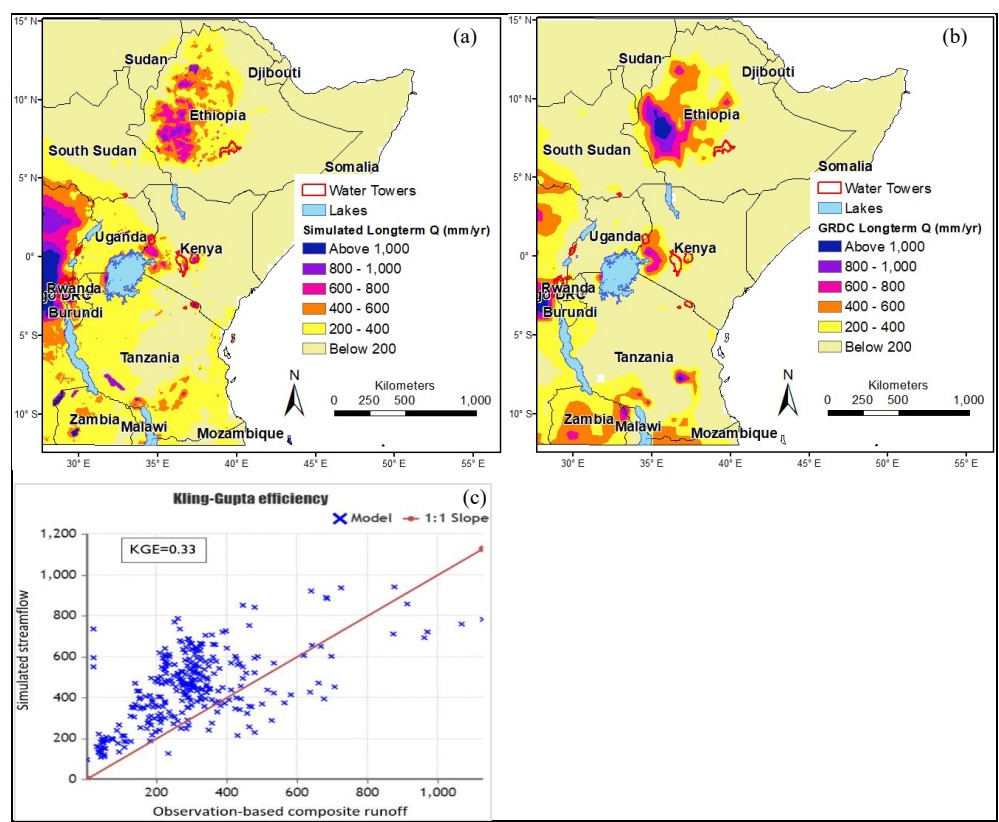

**Figure 8. Spatial distribution of streamflow in the East African region:** (a) **Simulated streamflow,** (b) **Observation-based composite runoff, and** (c) **Kling-Gupta efficiency calculation**

The water yield was observed to be relatively more sensitive to climate changes (i.e. P and PET) than land-use changes within the selected East African water towers. However, a closer look at the regions surrounding the selected water towers revealed that the effects of land-use changes have greater impacts on water yield outside the water towers boundaries (Fig 9). An example is on the Eastern side of Mt Elgon where there was a major reduction in water yield especially in the periods of 2001-2010 and 2011-2019, (Fig 9 Row no 3, column B). Climate changes showed a reduction of water yield in seven water towers in the periods of 1991-2000 and 2001-2010. However, in the years of 2011-2019, climate changes triggered increased water yield in seven water towers (Fig 10). The climate changes in Mt Elgon resulted in a consistent increase in water yield, while a consistent decrease was inferred for Mt Kilimanjaro.

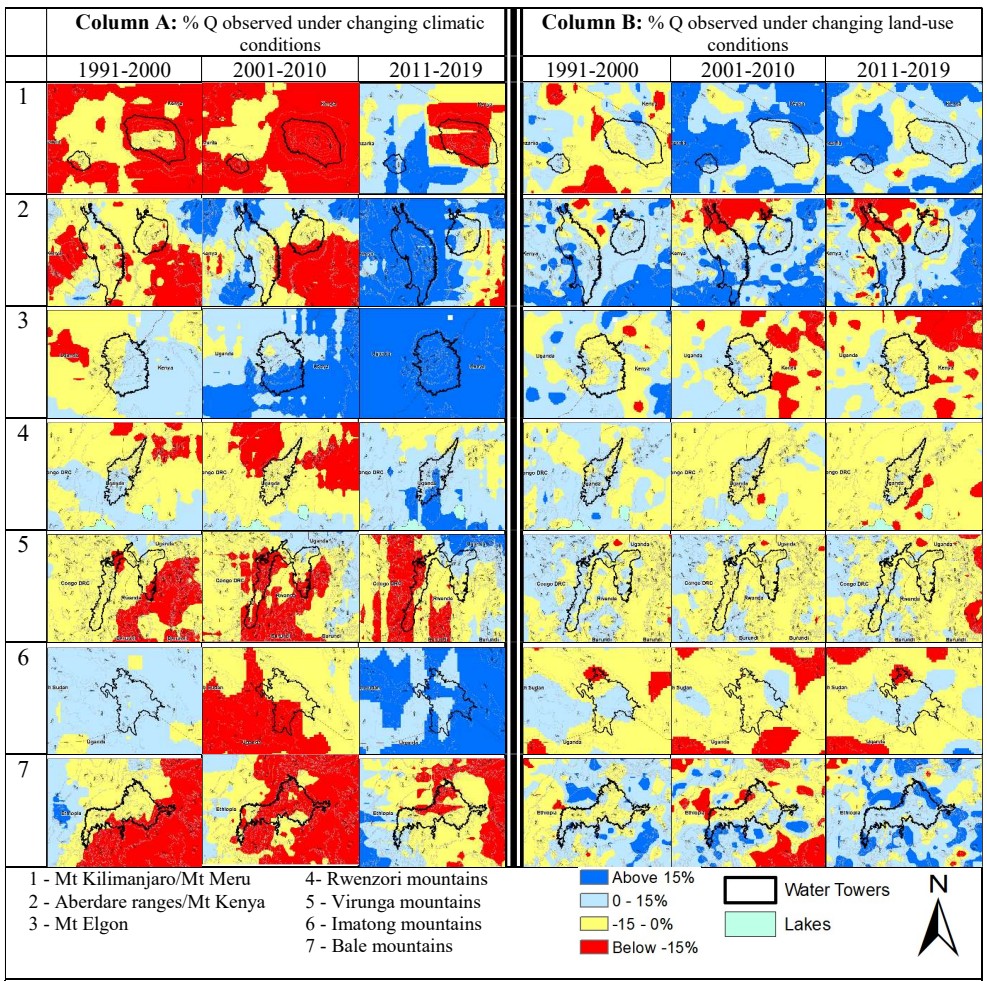

**Summary:**

**1. Mt Kilimanjaro/Mt Meru:** In column A the changes are observed both inside and outside water towers. In column B extreme changes are observed outside the water towers.

**2. Aberdare ranges/Mt Kenya:** In column A, major reductions were observed on the south-eastern side. In column B, major reductions were observed on the north-western side.

**3. Mt Elgon:** In column A, increases observed inside water towers. In column B, major decreases outside the water tower. No major changes inside water towers.

**4. Rwenzori mountains**: In column A, major changes inside the water tower. No major changes inside the water towers in column B.

**5. Virunga mountains:** In column A, there are major changes that also affect water yield inside the water tower. In column B, no noticeable changes inside the water tower. But extreme reductions were observed outside the water tower.

**6. Imatong mountains:** In column A, noticeable changes inside the water tower. In column B, no obvious changes inside the water towers, but extreme changes outside the water tower.

**7. Bale mountains:** In column A, there are apparent changes inside the water tower. In column B, no major changes inside the water tower.

**Figure 9. Effects of Land-use and Climate change on Water yield (Q)**

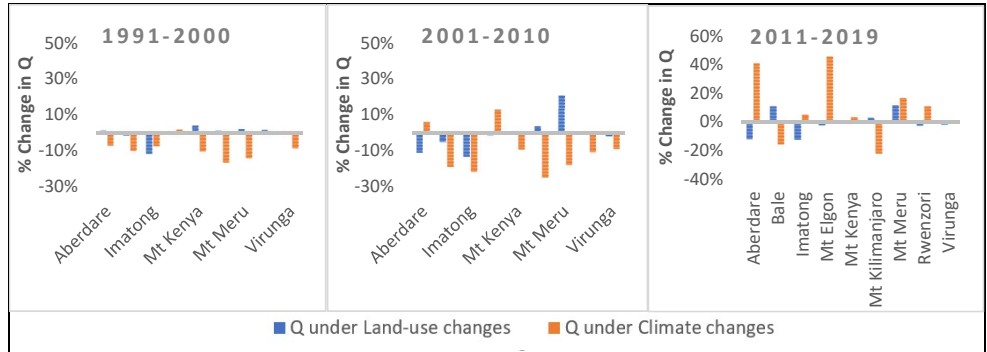

**Figure 10. The effects of land-use and climate changes on water yield**

The analysis of vertical deviations revealed downward shifts and upward shifts from the Budyko curve in the different water towers. The vertical deviations (d) ranged from negative (-)0.05 to positive (+)0.02 (Fig 11a). There
were no vertical deviations observed in Mt Elgon and Imatong mountains indicating that the values observed (between 1991 and 2019) were approximately close to those predicted by the Budyko curve. The elasticity (e) values ranged from 0.49 to 17.6 with most of the water towers recording lower elasticity values as shown in Fig 11b. The higher elasticity (e) values were observed in Mt Kenya water tower in the years of 1991-2000, Bale mountains in the period of 2011-2019, and Aberdare ranges in the periods of 1991-2000 and 2001-2010.

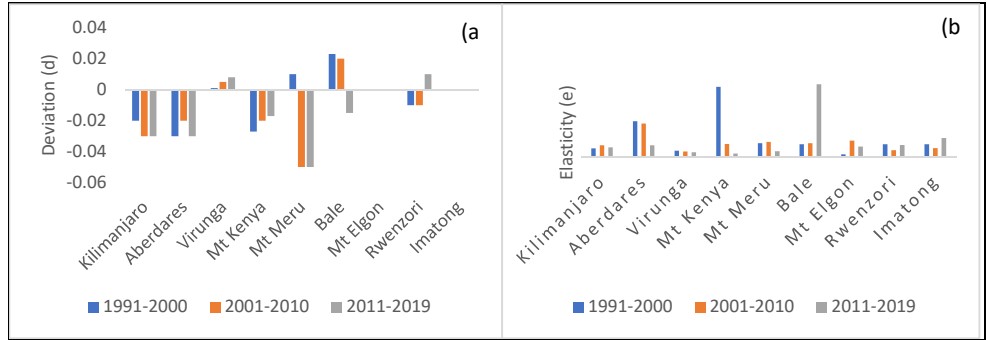

**Figure 11. The Deviation from the Budyko curve (a) and Elasticity (b) in the Water Towers**

The horizontal shifts either to the left or to the right relative to the dryness index (DI) were observed in the water towers. It was observed that 7 out of the 9 water towers plotted in the left (i.e. DI values less than 1, towards humid conditions) (see Fig A4 a-i in the appendix). However, 2 of the water towers (i.e. Mt Meru and Bale mountains)
plotted more towards the right (i.e. DI values greater than 1). Mt Meru seems to have shifted from warmer to humid conditions in the period of 2011-2019 as shown in Fig A4.

## 4. Discussion

The sensitivity analysis revealed differences brought about by climate changes and land-use changes on water
yield within and outside the water towers. Within the water towers, water yield was more sensitive to climate changes compared to land-use changes. Outside the water towers, the water yield was observed to be more sensitive to land-use changes than climate changes. This study, therefore, suggests that the direct anthropogenic influences have a stronger impact outside the water towers. However, the Budyko metrics analysis revealed vertical deviations (d) from the Budyko curve. According to Creed et al. (2014), these deviations indicate the presence of
anthropogenic effects within the water towers. In some of the water towers such as Mt Elgon and Imatong mountains, no vertical deviations were observed indicating that the changes in the two water towers can majorly be associated with naturally occurring oscillations.





Lower elasticity values were also observed in most of the water towers. Low elasticity indicates broad ranges in the evaporative indexes (EI) compared to Dryness indexes (DI) which further proves the presence of anthropogenic influence within the water towers. According to Creed et al. (2014), elasticity can be used as a measure of resilience. Elastic catchments are expected to plot along the Budyko curve (i.e. resilient to climate changes) while

inelastic catchments (non-resilience to climate changes) would deviate from the Budyko curve.

The horizontal shifts of the water towers either to the left or to the right relative to dryness index (DI) is an important indicator of the behavior of the water towers towards warmer conditions or humid conditions. These horizontal deviations reflect a change in the climatic conditions specifically temperature and precipitation (Creed and Spargo, 2012a). This study observed that the majority of the water towers (7 out of 9) plotted within humid

conditions (i.e. DI <1). On the other hand, two of the water towers (i.e. Mt Meru and Bale mountains) demonstrated warmer conditions (i.e. DI >1). One major observation is that water towers in Eastern Africa seem to shift towards the left, an indication of the increased humid conditions especially in the period of 2011-2019. This includes Mt Meru which shifted from warmer conditions observed in 1991-2000 and 2001-2010 to humid conditions in the years of 2011-2019.

A gradual increase in PET was observed in all the water towers. This indicates that the atmospheric demand is rising, an important signal of temperature increases in the East-African region. The effects of increasing temperature have already been identified to have decreased the surface area of glaciers by 80 % in East African water towers (EAC et al., 2016), affecting runoff and water resources downstream. According to Niang et al. (2015), the temperature in Africa is projected to rise faster than the rest of the world, which could exceed 2°C by

the mid-21$^{st}$ century and 4$^\circ$C by the end of the 21$^{st}$ century. Therefore, the water towers are under pressure from climate changes and PET is proving to be an important climate driver influencing water availability in the region. The mountainous forest ecosystems located in drier environments (such as Mt Kilimanjaro, Mt Meru, Mt Kenya, and Aberdare ranges) are important rainfall regions as they receive relatively higher rainfall than the adjacent areas. This ensures water availability in the adjacent lowlands in the arid and semi-arid (ASAL) regions.

The simulated evapotranspiration (ETa) and water yield (Q) revealed longitudinal differences with low to high values ranging from East to West. A related pattern on climate varying across East Africa from arid conditions in the east to more humid conditions in the west was also observed by (Daron, 2014). However, the individual water towers revealed independent variations that do not follow the longitudinal pattern. For instance, the higher mean annual water yield was observed in Mt Kilimanjaro despite being located in the drier environment on the Eastern

side. This emphasizes the importance of elevated forested areas in ensuring water availability in semi-arid areas. The extreme opposite trends observed in water yields from the different water towers confirm a strong variation in the climatic patterns. For instance, while there was a consistent increase in annual mean water yield in Mt Elgon, the opposite was true in Mt Kilimanjaro where a steady decline in water yield was observed.

The Budyko framework is a suitable approach for analyzing the partitioning of rainfall into precipitation and water

yield. The framework gives the possibility for analyzing the combined effects or separating the effects of climate and land-use changes on water yield. In this study, the spatial pattern of the simulated streamflow in the Budyko framework closely resembles the pattern observed in the GRDC composite runoff. The Kling-Gupta efficiency test revealed positive values, KGE=0.33 which are seen as "good" model performance (Knoben et al., 2019). Therefore, the Budyko simulations in this study were considered acceptable. However, it should be noted that this

comparison is added for reference only and should not be seen as validation. This is because, the Global composite runoff (Fekete et al., 2002) is not a strictly observational dataset, and it is used here as the "best estimate" available for long-term estimates of streamflow in the East African region. The fact the Budyko framework uses data and parameters that are easily measurable at a regional scale makes it a suitable approach for regions such as East Africa where there is a gap in the availability of detailed and quality local climatic data.

The major reference period for this study was the 1981-1990 period based on the CHIRPs rainfall with precipitation data beginning 1981 onwards. Further research using a reference period earlier than 1981 would help to strengthen the findings of this study especially after the evidence of shifts towards wetter conditions in all the water towers. Further studies on human-water interactions are also recommended to understand in detail the dynamics and co-evolution of coupled human-water systems.

## 5. Conclusions

Climate changes (i.e. changes in precipitation and potential evapotranspiration) have a relatively large effect on water yield within the East African water towers. The effects of land-use changes on water yield are larger in the adjacent regions surrounding the water towers. The majority of East-African water towers are under pressure from human influences both within and outside the water towers. Generally, the patterns in water yield showed a strong longitudinal difference (East to West), though the elevation is a key factor that ensures the generation of water in the water towers located in drier environments. A hydroclimatic phenomenon is occurring in the East-African region as the water towers show a strong shift towards wetter conditions (especially in the period of 2011-2019) while at the same time, the atmospheric demand is gradually increasing. Given that majority of the water towers were identified as non-resilient to changes, it means there are greater possibilities of extreme variations in water yield under changing climatic conditions. The Budyko framework provides a suitable approach especially for regions that lack detailed and quality data.

## 6. Appendices

### A. Extended Figures and Tables

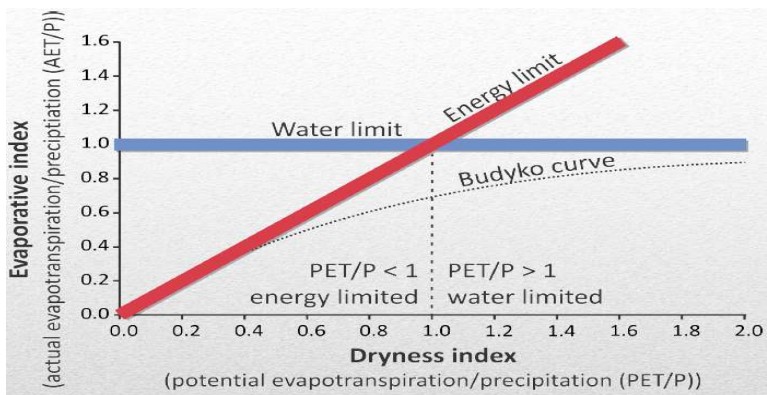

**Figure A 1. The Budyko Curve: Water limit (ETa=P); a site cannot plot above the blue line unless there is input of water beyond precipitation. Energy limit (ETa=PET); a site cannot plot above the red line unless precipitation is being lost from system by means other than discharge (adapted from Creed and Spargo, 2012a).**

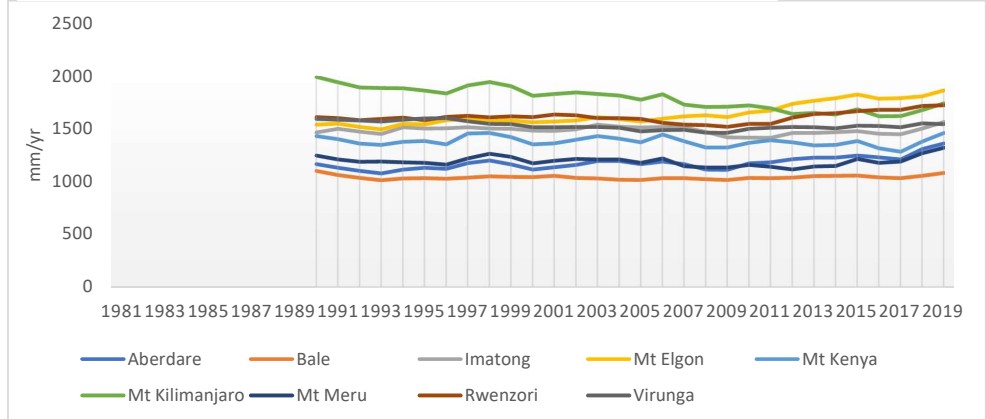

**Figure A 2. The 10-year Moving averages of Annual mean precipitation in the different water towers**

**Higher levels of annual precipitation were observed in 1990-1992, 1998-1999, and 2013-2015. Lower levels of annual precipitation were observed in 1987, 1995-1996, 2004-2005, and 2017. Mt Kilimanjaro was observed to have a consistent decline in annual mean rainfall between 1981 and 2017. Mt Elgon water tower recorded a consistent increase in annual mean rainfall during the study period.**



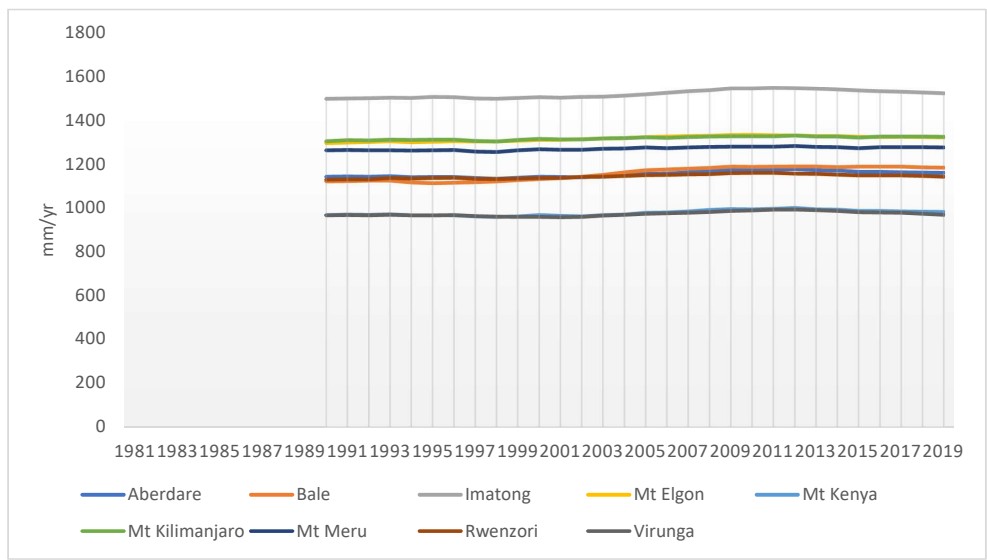

**Figure A 3. The 10-year Moving averages of Annual mean PET in the different water towers**





**Table A 1. Summary of simulated evapotranspiration (ET) and water yield (Q) given the precipitation (P) between 1981 and 2019**

| Mountain Ecosystems | PE | FSC | A | 1981-1990 | | | | | 1991-2000 | | | | | 2001-2010 | | | | | 2011-2019 | | | | |
|---|---|---|---|---|---|---|---|---|---|---|---|---|---|---|---|---|---|---|---|---|---|---|---|
| | | | | P | ET | Q | Q | RR | P | ET | Q | Q | RR | P | ET | Q | Q | RR | P | ET | Q | Q | RR |
| | | | | | | | FR | % | | | | FR | % | | | | FR | % | | | | FR | % |
| Units | m | m | km² | mm/yr | mm/yr | | | | mm/yr | mm/yr | | | | mm/yr | mm/yr | | | | mm/yr | mm/yr | | | |
| Mt Kilimanjaro | 5,895 | 2000 | 1513.0 | 1989.1 | 1056.3 | 919.1 | 0.029 | 46 | 1811.2 | 1019.2 | 779.3 | 0.025 | 43 | 1720.7 | 980.7 | 727.9 | 0.023 | 42 | 1751.4 | 991.4 | 748.0 | 0.024 | 43 |
| Mt Kenya | 5,199 | 2000 | 3298.3 | 1429.0 | 867.5 | 551.5 | 0.017 | 39 | 1350.5 | 828.5 | 512.7 | 0.016 | 38 | 1365.0 | 861.7 | 494.6 | 0.016 | 36 | 1466.4 | 884.0 | 573.7 | 0.018 | 39 |
| Mt Elgon | 4,321 | 2000 | 2548.0 | 1538.4 | 1051.3 | 480.8 | 0.015 | 31 | 1560.0 | 1068.7 | 483.6 | 0.015 | 31 | 1653.8 | 1110.2 | 535.5 | 0.017 | 32 | 1856.9 | 1159.5 | 686.9 | 0.022 | 37 |
| Aberdare Ranges | 3,999 | 2100 | 6671.8 | 1161.0 | 804.3 | 353.0 | 0.011 | 30 | 1110.2 | 772.1 | 335.2 | 0.011 | 30 | 1169.3 | 804.7 | 360.9 | 0.011 | 31 | 1352.8 | 869.7 | 479.1 | 0.015 | 35 |
| Rwenzori Mountains | 5,109 | 2000 | 1465.0 | 1609.5 | 934.1 | 653.6 | 0.021 | 41 | 1610.1 | 931.1 | 656.1 | 0.021 | 41 | 1546.0 | 944.7 | 580.4 | 0.018 | 38 | 1705.7 | 970.2 | 710.0 | 0.023 | 42 |
| Mt Meru | 4,565 | 2000 | 225.5 | 1244.4 | 898.2 | 330.6 | 0.010 | 27 | 1171.1 | 866.2 | 290.2 | 0.009 | 25 | 1156.1 | 803.7 | 338.0 | 0.011 | 29 | 1340.6 | 894.8 | 427.2 | 0.014 | 32 |
| Virunga Mountains | 4,507 | 2000 | 5288.2 | 1591.8 | 847.6 | 737.1 | 0.023 | 46 | 1512.4 | 833.5 | 672.0 | 0.021 | 44 | 1498.2 | 861.4 | 629.0 | 0.020 | 42 | 1523.9 | 850.0 | 666.8 | 0.021 | 44 |
| Bale Mountains | 4,337 | 2600 | 5030.7 | 1099.4 | 755.3 | 343.3 | 0.011 | 31 | 1040.5 | 741.9 | 299.7 | 0.010 | 29 | 1034.4 | 755.2 | 279.6 | 0.009 | 27 | 1071.3 | 733.5 | 338.4 | 0.011 | 32 |
| Imatong Mountains | 3,187 | 2000 | 396.7 | 1464.9 | 1084.4 | 369.2 | 0.012 | 25 | 1482.5 | 1102.1 | 368.8 | 0.012 | 25 | 1416.7 | 1097.1 | 305.7 | 0.010 | 22 | 1576.6 | 1145.6 | 416.8 | 0.013 | 26 |

*Note:*

*P = precipitation*
*ET = actual evapotranspiration*
*Q = water yield*
*RR = Runoff Ratio*
*m = meters*
*km² = square Kilometers*
*FR = Flow rate in m³/s per km²*
*PE = Peak Elevation*
*FSC = Foot slope contour*
*A = Area*

**Figure A 4. Graphical representation of the baseline Budyko curve (estimated for 1981-1990 and the trends of water towers in the different years**



## 7. Data availability

All climatic data used in this study (i.e. (P, PET, and NDVI) are publicly available. Precipitation data (P) was downloaded from Climate Hazards Group Infrared Precipitation with Stations (CHIRPS-v2) https://www.chc.ucsb.edu/data/chirps (last accessed 31 July 2020). Potential Evapotranspiration (PET) data was
downloaded from the CRU database https://crudata.uea.ac.uk/cru/data/hrg/ (last accessed 22 July 2020). Normalized Difference Vegetation Index (NDVI) was sourced from Global Inventory Monitoring and Modeling System (GIMMS) Third Generation (3 g) Advanced Very High-Resolution Radiometer (AVHRR) sensor onboard the National Oceanic and Atmospheric Administration (NOAA) satellites https://ecocast.arc.nasa.gov/data/pub/gimms/3g.v1/ (last accessed 12 July 2020). The water towers analyzed data
sets are summarized in Table A1 for each water tower and the full analysis is available in the SESAM project SharePoint. The data can be provided upon request from the 1st author (charles.wamucii@wur.nl).

## 8. Author contributions

CNW and AJT designed the study. CNW performed the analysis and wrote the manuscript. All authors helped with the interpretation of the results and commented on the manuscript.

## 9. Competing interests

The authors declare that they have no conflict of interest.

## 10. Acknowledgments

This research was made possible by Wageningen University through the Scenario Evaluation for Sustainable Agro-forestry Management (SESAM) project that was funded by its Interdisciplinary Research and Education Fund
(INREF).

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
