# Peer review of "Land-use and climate change effects on water yield from East African Forested Water Towers"

_Hydrology and Earth System Sciences, 2021_

## Referee Comment (RC2)

[referee-annotated manuscript omitted]

---

## Author Comment (AC1)

**RC3: 'Comment on hess-2021-151', Steve Lyon,**

This study investigates climate and human impacts on water towers in East Africa. The analysis is conducted in a Budyko framework. The target region is often considered vulnerable to changes in water resources making this investigation warranted and the result likely informative. Overall, the study is well-conceived. However, I feel there are some considerable limitations in the structure of the presentation. Further, some of the mechanistic interpretations are not fully supported given the potential of confounding impacts and potential uncertainty in data and analysis.

We thank the reviewer for the constructive comments. We have considered all comments and suggestions in our revised manuscript.

1. In the introduction (especially around P2, L15-L20), I would expect to see some more consideration of the strengths and weaknesses of various approaches for assessing climate and land-use change on water resources. Do we have some results or previous work that are relevant for the region? What is the motivation for selecting the Budyko approach over other approaches? There is not many reviews of current science offered up in the introduction. This should be expanded to help the reader understand the motivations for the current study and approach.

**Answer:** We thank the reviewer for the remark. We have reviewed additional literature. We have also described the strengths and weaknesses of various approaches for assessing climate and land-use change on water resources. A justification has also been added on the motivation for selecting the Budyko approach. The following text is an extract from the revised manuscript describing the strengths and weaknesses of various approaches and justification for selecting the Budyko framework:

*...Various approaches have been used for studying the effects of climate and land-use changes on streamflow. Jiang et al. (2015) categorized such methods into two: (a) deterministic rainfall-runoff models and (b) statistical methods. Ma et al. (2014) combined the two categories by running rainfall statistics and recorded land-use change patterns in reverse order in calibrated process-based models. Dey and Mishra (2017) reviewed the existing approaches and categorized these approaches into four categories; (i) experimental approach e.g. paired catchment method (Bosch and Hewlett, 1982), (ii) hydrological modeling e.g. SWAT (Tech, 2019), (iii) conceptual approaches e.g. Budyko approach (Budyko, 1974), and (iv) analytical approaches e.g. climate elasticity method (Schaake, 1990).*

*Generally, the different approaches can be grouped into modeling and non-modeling (conceptual) approaches (Marhaento et al., 2017). The advantage of modeling approaches is that the results are more reliable (Booij et al., 2019). However, the challenges of modeling approaches (e.g. SWAT) are that the underlying processes must be explicit, require complex and multiple data inputs, and a time-consuming calibration and validation (Zhang et al., 2012). Application of modeling approaches is therefore limited to relatively small watersheds where detailed streamflow observations are available or in watersheds that are well monitored with extensive, long-term available data on vegetation, soil, topography, land use, hydrology, and climate (Wei and Zhang, 2011). Non-modeling approaches such as Budyko conceptual frameworks require fewer data, hence flexible in*

*their application from small to large study areas and generally give logical primary results (Booij et al., 2019; Marhaento et al., 2017). These primary results can be crucial for data-limited regions such as East Africa and can form the basis for detail hydrological studies. In this study, we selected the Budyko framework assuming steady-state to analyze the impact of land-use and climate changes on water yield for the selected forested water towers of East Africa...*

*....Generally, the Budyko framework, either in the original format (i.e. steady-state) or in the modified format (i.e. non-steady-state conditions) is a quick first-order tool for estimating precipitation partitioning into evaporation and water yield (Mianabadi et al., 2020; Teng et al., 2012; Zeng et al., 2020)...*

In addition, the lack of framing the study in a research question or a hypothesis is a major weakness. The result is that the study is some exploration of data that does not seem to address a problem or help advance the science. Such exploration ("can-we-do-it" type of work) is fine for a technical report but more would be needed for publication in a peer-reviewed journal. I am confident the authors can put this study in a research framework and present a clean and testable hypothesis or a some societally relevant research question.

**Answer:** We agree with the reviewer and we have now included a focused research question:

"What are the effects of climate and land-use changes on water yield for the selected forested water towers?"

We tested the following hypothesis:

In areas considered as pristine or protected zones (i.e. high elevated forested areas), with AI≥0.65, changes in water yield would majorly be attributed to climate changes and negligibly due to land use/cover changes. The high elevated forested areas would then be expected to fall on the reference Budyko curve over the study period.

2.  The study mixes direct observation data interpolated across sites and remotely sensed data at various scales. I'm wondering if there is any potential impact of the various assumptions and approaches in each dataset? Synthesizing across various approach can often compound huge amounts of uncertainties and errors as we build composite analysis (in space and time). How has uncertainty been considered in your analysis and what role would data error have on your results/interpretation? Some consideration and discussion of uncertainty impacts must be presented to help the reader understand how robust the findings are in this study. This should be fairly straight forward given how the water balances were constructed using 100 random points. Perhaps perform a re-selection of random point and assess the difference or use some sort of calibration/validation approach on a sub-division of the 100 points (like a boot strap).

**Answer:** We thank the reviewer for this comment. A discussion on uncertainties has now been included in the revised manuscript. The feedback by the reviewer on water balance construction using 100 random points made it clear that clarification and reorganization of the manuscript is required. We used the 100 random points to develop the Budyko curves

and not for constructing water balances. The Budyko equation was applied for the whole region to simulate ET and Q. To increase clarity, we have introduced sub-sections in the revised manuscript. The sub-section that describes the use of 100 random points is called:

**...Developing the Budyko curves**

*To develop Budyko curves that are representative of the selected forested water towers, 100 random points were generated for each of the water towers in ArcGIS. The random points were used to extract values from raster P, PET, and ET grids into Spreadsheet for developing the Budyko curves. For maximum representation, the minimum allowed distance between the random points was set to 100 meters. The random points generated were assigned the respective values of PET, ET, and P using the Extract Multi Values to Points tool in ArcGIS. The Evaporative index (EI) values -calculated as a ratio of ET and P, and Dryness index (DI) values -a ratio of PET and P were used to draw the Budyko curves. In this study, the Budyko curve for the 1981-1990 period was used as the reference condition for the water balance, to effectively assess the trends in the succeeding periods of 1991-1990, 1991-2000, 2001-2010, and 2011-2019...*

For calibration/validation purposes, the simulated water yield (Q) was evaluated against observation-based runoff. This is now provided in a subsection called:

**...Comparison of simulated streamflow with observation-based runoff**

*The simulated streamflow of the water towers was compared with composite runoff data downloaded from the Global Runoff Data Centre (GRDC). The composite runoff fields, developed through combining observed river discharge information with a climate-driven water balance model, provide the "best estimate" of terrestrial runoff over large domains (Fekete et al., 2002). A total of 312 points above 2000 meters above sea level, which is the focus of this study (i.e. elevated water towers), were randomly generated in ArcGIS. For maximum representation, the minimum allowed distance between the random points was set to 100 meters. The selected random points and their respective values of simulated streamflow and composite runoff were compared...*

3. Further, I am not sure about the 100 random points in the methodology. Why was this done? Is it just too difficult to define the spatial extent of the water towers (which would allow using all the spatial data in the area)? Seems there would be some value in conducting this experiment at various elevations to assess the impact of elevation (as temperature proxy) on the results. Please outline why the method of 100 random point was selected and what the impacts would be on the results relative to another method.

**Answer:** We thank the reviewer for the remark. As explained in the above response, we agree a clear presentation in the manuscript is needed which we have now provided by organizing the manuscript into sub-sections. In fact, we did not use the 100 random points to construct the water balances. Rather, the water balances were constructed using all spatial data generated (gridded data for P, PET, NDVI etc). Actually, the water balances were constructed with gridded data for the entire East-African region, before selecting the data for our focus regions by masking out the spatially delineated extents of the water towers.

The 100 random points were only used to extract data from raster outputs with the sole purpose of developing Budyko curves - which was done using a spreadsheet model (MS Excel). The 100 points were generated for each of the water towers.

We agree with the reviewer's suggestion to include the elevation element in the analysis – this has been done in the simulation of longterm actual evapotranspiration (ET) and water yield (Q), potential evapotranspiration (PET), and precipitation (P). As a result, we have added the following figure as Figure A4 in the revised manuscript.

[Figure]

*Figure 1: The Impact of elevation on hydroclimatic conditions in the East African region*

*...The elevation influences hydroclimatic conditions in the East African region. The average atmospheric demand increases with a reducing elevation gradient. There is a steady increase in P, ET, and Q as elevation increases. For regions above 2000 m a.s.l, the precipitation exceeds potential evapotranspiration as shown in Fig 4A. This demonstrates the importance of the elevated humid zones in generating and sustaining water yield to the adjacent lowland areas...*

4. There appears to be a large amount of mechanistic speculation on why points depart from the Budyko curve. There has been ample research over recent decades explaining how we can see variations along and from the curve. Further, many different explanations have been offered as to why catchments would deviate from theoretical curves with time. Could you outline some motivation for how you can be certain you are isolating mechanisms with your analysis? We would anticipate much interaction and coupled response that could be masked in the movement of points in Budkyo space (see van der Velde et al., 2014). It i likely that this lack of consideration of complexity relates back to the weakness and lack of thorough literature review seen in the introduction.

**Answer:** We thank the reviewer for this comment. We have provided more details regarding our basis of interpretation in the revised manuscript. We have added the following text in the methodology section.

*....In this study, we used the Budyko framework and two recently introduced Budyko metrics (deviation and elasticity) (Creed et al., 2014b) to study the changes in the water yields. Similar methodologies were adopted by Helman et al. (2017) to determine the resilience of forested catchments and Sinha et al. (2018) to understand the involvement of anthropogenic stress and climatic variance on the partitioning of precipitation. Based on these studies, catchments can be assumed to shift predictably along the Budyko curve. This acts as a basis for interpreting the vertical and horizontal deviations as a result of changes in climate and anthropogenic effects. The elasticity is defined as a measure of a catchment's ability to maintain hydroclimatic conditions as the climate varies. In contrast to other studies using the Budyko framework to look at different drivers of change, we use Budyko-derived data rather than observations. Therefore, the deviations are thereby constructed and presented as a way to visualize the results. Beyond the maps and graphs presented following the Budyko equation we further illustrate the movement of water towers within the Budyko space...*

5. Along these same lines, what role would other factors such as CO2 increase and/or human alteration to water usage have in these regions? I could envision shifts in water cycling due to an intensification of plant activity through increased NPP or agricultural intensification. Warmer and CO2 richer climates could behave differently. Further, how much pumping and/or movement through irrigation schemes takes place in some of these systems? I understand they should be pristine or high-elevation forest without impact, but are they really without abstraction or other anthropogenic impacts?

**Answer:** We thank the reviewer for this comment. First, we have made it clear in the revised manuscript that the water towers are the high elevated forested areas, that are humid (i.e. aridity index AI≥0.65) and are considered pristine (under protection). The forests are under a continuous forest block from the footslope to the mountain peak. Based on our hypothesis, we are investigating whether such regions' changes in water yield would majorly be attributed to climate changes and negligibly due to land use/cover changes. The high elevated forested areas would then be expected to nicely fall on the reference Budyko curve over the study period.

Given the framework used, we only look at human alterations in the form of land cover changes without going into details about the type and sources of effects e.g. effect of $CO_2$ increase, irrigation schemes/ agricultural intensification, etc. We believe the impact of warming is captured in the potential evapotranspiration. In fact, by looking at NDVI we are accounting for possible $CO_2$ effects on vegetation growth (though not on stomatal opening and ET directly).

That said, in the discussion section, we have added details of other factors that would also affect the results. We have reviewed additional publications that looked at $CO_2$ and human alteration of the water usage and hypothesize potential effects that could be contributing to the observed changes. This can further inform areas of further research. The following relevant text has been added in the discussion section:

*...This study focussed on the role of water towers in the supply of blue water to downstream parts of the watershed. These findings can be improved further by studying the role of water towers in the supply of green water (i.e. the role of water towers in regional rainfall*

*recycling). We also recognize other factors that may influence the results in this study. For instance, increasing atmospheric $CO_2$ concentrations may affect terrestrial water cycling through changes in climate and changes in transpiration (i.e. stomatal conductance) (Frank et al., 2015; Huntington, 2008; Mamuye, 2018). We also note that if $CO_2$ leads to higher NDVI, then this effect is accounted for in our modeling approach. Some studies have reported that NDVI linear trends can be linked to increasing $CO_2$ levels (Krakauer et al., 2017; Yuan et al., 2017). However, further investigations are recommended. Other factors that may affect our results include the human alteration to water usage. Kiteme et al. (2008) reported unregulated abstraction of water in the upstream of Mt Kenya water tower leading to hydrological droughts in the downstream. Intensification of irrigated agriculture and a growing human population was reported at the foot slopes of the water towers (Liniger et al., 2005; Ulrich et al., 2012). The effects of anthropogenic presence at the foot slope of the water towers have not been accounted for and further studies are required to understand how humans affect the pristine/protected water towers...*

6.  In general, the results as presented are dense and not easy to follow. Read things a few times and not sure I can understand all the nuance of what is being shown here due to how things are being presented. This is not helped by poorly constructed figures with overlapping number, limited axis labels, and multiple colors to track. A major effort to organize the results into a concise section is required. Start by group the various results into sub-sections and cleaning up the figures. Structuring this section could also be aided by a more thoughtful research question or hypothesis setup. Then the results could be organized into how they answer the research question(s).

**Answer:** We thank the reviewer for this feedback. The results section is now organized into sub-sections. The figures have been cleaned up to remove the overlapping numbers and shading also recommended by Reviewer RC2. We have constructed the research question which has also been reflected in the results and discussion sections. The results section is now organized according to the following sub-sections:

- *Climate characteristics over the period 1981 – 2019 (Precipitation and Potential evapotranspiration)*
- *Land cover characteristics over the period 1981 – 2019*
- *Simulation of Evapotranspiration*
- *Simulation of Water Yield*
- *Comparison of simulated streamflow with existing runoff data*
- *The effects of land use and climate changes on water yield*
- *Analyzing the water towers in the Budyko space*

All figures are cleaned as demonstrated in the following example:

[Figure]

A revised figure, with no overlapping numbers. X-axis label now included

Old figure with overlapping numbers, with no X-axis label

7. The discussion section is lacking rigor. At best it repeats the results with more interpretation. I miss a connection to the literature and how the results help inform and advance the science. Also, what are the strengths and limitations of the approach considered and how do these impact interpretation? Could not see what value the discussion added to the paper overall. Rather, it felt like the results were being explained again and the assumptions behind interpretation being ignored. Lastly, while there are no rules, the length of the discussion is rather short relative to the length of the results presented. In my experience, that can be indicative of a study that is exploring data rather than an experiment to test a hypothesis.

**Answer:** We thank the reviewer for bringing this to our attention. Together with comments from the reviewer (RC1), we have revised the discussion section to ensure the results are better interpreted, linking to existing literature, and a discussion on uncertainties/limitations. The discussion section has been revised and expanded as follows:

*Discussion*

*We found that within the water towers, water yield was more sensitive to climate changes than to land-use changes. In contrast, outside the water towers, the water yield was observed to be more sensitive to land-use changes than to climate changes. This suggests that anthropogenic influences are relatively higher outside the water towers. Contrary to our expectation, our analysis showed that most of the water towers (i.e. 7 out of 9) did not plot on the reference Budyko curve over the study period. This is a relevant finding since all water towers were considered pristine and protected. Only two water towers, Mt Elgon and Imatong mountains showed no deviations from the reference Budyko curve. Generally, our investigation highlights the importance of elevated water towers in a semi-arid region in the generation and supply of water to adjacent lowland areas. The forested water towers located in drier environments (such as Mt Kilimanjaro, Mt Meru, Mt Kenya, and Aberdare ranges) are important rainfall regions as they receive relatively higher rainfall than the adjacent areas. This ensures water availability in the adjacent lowlands in the arid and semi-arid (ASAL) regions.*

*Our results indicate that changes in precipitation and potential evapotranspiration are the major determinants of blue water availability from high elevated forested water towers in*

*the East African region. Related observations have been made - that climate changes in Africa have a relatively higher impact on water yield compared to other drivers such as land-use changes (Alcamo et al., 2007; Niang et al., 2014). However, the lack of evidence of sensitivity to land-use changes within the water towers themselves may be linked to existing institutional arrangements. We presume that the results would be different if such rules would be relaxed. That said, the movement of water towers in the Budyko space revealed that anthropogenic influence within the water towers cannot be ruled out. Our analysis revealed vertical deviations (d) from the Budyko curve for 7 out 9 forested water towers. According to Creed et al. (2014), these vertical deviations may indicate the presence of anthropogenic effects within the water towers. The two water towers where no deviations were observed (i.e. Mt Elgon and Imatong mountains), indicate that the hydroclimatic conditions in the study period did not vary much from the reference conditions of 1981-1990 and any changes in water yield in the two water towers can largely be associated with climatic changes in P and PET.*

*Moreover, the lack of deviations in the two water towers may indicate the resilience of forested regions (i.e. adaptable nature of forests) as described in (Creed et al., 2014; Helman et al., 2017; Van der Velde et al., 2014). Such resilience (measured as elasticity) could be a key factor in forested water towers indicating their ability to resist change or bouncing back to their initial natural conditions, hence plotting along the reference Budyko curve. Long-term adaptations of forests have been achieved by trees even in the most water-limited forests (Helman et al., 2017). However, our investigations on elasticity (that refers to the degree of initial change using 1981-1990 as the reference period) did not support the above science as lower elasticity values were observed in most of the water towers. Given that low elasticity indicates broad ranges in the evaporative index (EI) compared to the dryness index (DI), this may further indicate the presence of anthropogenic influence within the water towers. According to Creed et al. (2014), elastic catchments are expected to plot along the Budyko curve (i.e. high elasticity = resilient to climate changes) while inelastic catchments (i.e. low elasticity = non-resilience to climate changes) would deviate from the Budyko curve.*

*Further illustrations can be shown in the Budyko space based on the horizontal shifts relative to the dryness index (DI). The horizontal shifts are important indicators of the behavior of the water towers towards warmer or humid conditions. These horizontal deviations reflect a change in the climatic conditions specifically, temperature and precipitation (Creed and Spargo, 2012a). This study observed that the majority of the water towers (7 out of 9) plotted within humid conditions (i.e. DI <1). On the other hand, two of the water towers (i.e. Mt Meru and Bale mountains) demonstrated warmer conditions (i.e. DI >1). One major observation is that water towers in Eastern Africa seem to shift towards the left, an indication of the increased humid conditions especially in the period of 2011-2019. At the same time, a gradual increase in PET was observed in all the water towers. A climate shift to wetter conditions and simultaneous increases in regional temperatures have also been reported in the East African region and projected to increase by the end of 21st century (Giannini et al., 2018; Niang et al., 2014; Omambia et al., 2012).*

The effects of increasing temperatures have already been identified to have decreased the surface area of glaciers by 80 % in East African water towers (EAC et al., 2016), affecting runoff and water resources downstream. According to Niang et al. (2014), the temperatures in Africa is projected to rise faster than other parts of the world, which could exceed 2°C by the mid-21$^{st}$ century and 4°C by the end of the 21$^{st}$ century. Therefore, the water towers are under pressure from climate changes and PET is proving to be an important climate driver influencing water availability in the region. There are chances that the shifts to wetter conditions in the water towers may also be as a result of the extended impact of increasing PET on the El Nino-Southern Oscillation (ENSO), a phenomenon that influences precipitation in the East African region. Li et al. (2016) investigated annual flood frequencies, from 1990 to 2014, and observed upward trends that were linked to the ENSO phenomenon. Additionally, the shifts to wetter conditions also coincide with the recent reports on the 'rising lake levels phenomenon' in the Eastern Africa region (Chebet, 2020; Chepkoech, 2020; Patel, 2020; Wambua, 2020). We however do not believe we have the data to link the climatic shifts and 'swelling' of lakes to ENSO variations in our study which requires detail scientific investigations.

The simulated evapotranspiration (ET) and water yield (Q) revealed longitudinal differences with low to high values ranging from East to West. A related pattern on climate varying across East Africa from arid conditions in the east to more humid conditions in the west was also observed by Daron (2014). However, the individual water towers revealed independent variations that do not follow the longitudinal pattern. For instance, a higher mean annual water yield was observed at Mt Kilimanjaro despite being located in the drier environment on the Eastern side. This emphasizes the importance of elevated forested areas in ensuring water availability in semi-arid areas. For instance, in high elevated forested zones, the precipitation exceeds potential evapotranspiration, which ensures a surplus of blue water that eventually flows downstream.

The extreme opposite temporal trends observed in water yields from the different water towers confirm a strong variation in the regional climatic patterns. For instance, while there was a consistent increase in annual mean water yield at Mt Elgon, the opposite was true at Mt Kilimanjaro where a steady decline in water yield was observed. Our results further revealed that precipitation (P) is the dominant driver in the East African region. For instance, a consistent increase in Q at Mt Elgon coincided with a steady increase in land surface characteristics ($\omega$) as shown in Figure 3 C. Ideally, a reduction in Q would have occurred due to the increase in ET (associated with increases in land surface characteristics), but this was diffused by the increases in rainfall as shown in Figure 2 C. At Kilimanjaro water tower, a continuous reduction in Q coincided with a steady reduction in $\omega$. Again, an increase in Q would have been expected due to a decrease in ET. Therefore, precipitation is the dominant driver in the generation and supply of blue water from the forested water towers in the East African region.

As a first-order tool, the Budyko framework provides an important reference point for relating variations in water yield to variations in climatic conditions and catchment properties. In this study, the spatial pattern of the simulated streamflow in the Budyko framework closely resembles the pattern observed in the GRDC composite runoff. We however noted overestimation of water yield in the comparison. This type of observation

was also reported by (Teng et al., 2012), where the Budyko equation was found to overestimate water yield in drier regions. Moreover, other factors such as soil type, topography, seasonality, water storage, interception, etc were not accounted for in the quantitative framework which can affect the simulations in the selected forested water towers.

Canopy interception, for instance, plays an important role in the water balance of forested ecosystems as noted in several studies (Astuti and Suryatmojo, 2019; Gash et al., 1980; Teuling et al., 2019; Zimmermann et al., 1999). In their study, (Teuling et al., 2019) found many forested points to have average yearly evapotranspiration (ET) that exceeds the average potential evapotranspiration (PET). Van Dijk et al. (2015) opined that this is possible due to underestimation of evapotranspiration which was attributed to evaporation of interception water by energy not captured in the formulation of PET. The forest evapotranspiration paradox is further discussed in (Teuling, 2018). The correction of underestimation in (Teuling et al., 2019) indicates the need for long-term lysimeter observations for studies focussing on forested ecosystems. Availability of meteorological data in the upper slopes of the East African mountains is a big gap as the majority of meteorological observations are conducted below 1500 m a.s.l and most of the upper slopes data rely on extrapolation of hydrological analysis in the lowlands (Røhr and Killingtveit, 2003).

Local-based runoff measurements would have helped to interpret if there is indeed an overestimation in our study. That said, we observed positive KGE which indicates a "good" model performance (Knoben et al., 2019). Therefore, we considered the Budyko simulations as acceptable. However, it should be noted that this comparison is added for reference only and should not be seen as validation. This is because, the Global composite runoff (Fekete et al., 2002) is not a strictly observational dataset, and it is used here as the "best estimate" available for long-term estimates of streamflow in the East African region. The fact the Budyko framework uses lesser data and parameters that are easily measurable at a regional scale makes it a suitable approach for data-limited regions such as East Africa.

Besides the strengths in using the Budyko approach, uncertainties may exist which could have affected our results. The study used data from different datasets (CHIRPs, CRU, GIMMS/AVHRR) at various scales which could potentially affect results due to various assumptions and approaches in the processing of each dataset. For instance, the CRU dataset is fairly coarse and contains rather few observations in Africa. One substantial weakness of the current CHIRPS algorithm is the lack of uncertainty information provided by the inverse distance weighting algorithm used to blend the CHIRP data and station data (Funk et al., 2015). The overall NDVI3g uncertainty comes from spatial and temporal coherence variability which gives approximately an error of ±0.002 NDVI units. However, this NDVI error is considered low uncertainty hence applicable to study seasonal and inter-annual non-stationary phenomena (Pinzon and Tucker, 2014). Uncertainties may also arise in the general assumption that estimation of land surface characteristics ($\omega$) based on NDVI formulation provides values that represent integrated conditions for soil, topography, and climate seasonality. Some studies using various hydrological approaches have reported the significance of these factors in influencing catchment hydrology (Kirkby et al., 2002; Troch

*et al., 2013; Western et al., 2004; Woods, 2002). There is a need for more research to come up with methodological consistency in estimating ω parameters when using the Budyko framework. Although the focus of the study was in the elevated forested areas, empirical adjustment of the Budyko model may be needed to capture special features such as desert wadis in the application of the Budyko equation in the lowland areas.*

*We also recognize other factors that may influence the results in this study. For instance, increasing atmospheric $CO_2$ concentrations may affect terrestrial water cycling through changes in climate and changes in transpiration (i.e. stomatal conductance) (Frank et al., 2015; Huntington, 2008; Mamuye, 2018). We also note that if $CO_2$ leads to higher NDVI, then this effect is accounted for in our modeling approach. Some studies have reported that NDVI linear trends can be linked to increasing $CO_2$ levels (Krakauer et al., 2017; Yuan et al., 2017). However, detailed investigations are recommended within the East African region. Other factors that may affect our results include the human alteration to water usage. Kiteme et al. (2008) reported unregulated abstraction of water in the upstream of Mt Kenya water tower leading to hydrological droughts downstream. Intensification of irrigated agriculture and a growing human population was reported at the foot slopes of the water towers (Liniger et al., 2005; Ulrich et al., 2012). The effects of anthropogenic presence at the foot slope of the water towers have not been accounted for and further studies are needed to understand how humans living at the footslope of protected water towers affect the pristine conditions of the water towers at high elevations.*

*Notwithstanding these limitations, our study offers important findings on the sensitivity of water yield to climate and land-use changes and the importance of these water towers in the generation and supply of blue water to adjacent lowland areas. These results can be used by decision-makers, policymakers, stakeholders,  and scientists to emphasize the need to protect and conserve the high elevated forested areas in the region, particularly forest ecosystems above 2000 m a.s.l – where there is a surplus of blue water. The Budyko framework provides primary results that can inform detail hydrological assessments. For instance, our findings show that elevated forested water towers are important areas for maintaining high ET in the region. This finding can be explored further by studying the role of water towers in the supply of green water in the region (i.e. the role of water towers in regional rainfall/moisture recycling) (Ellison et al., 2017; Keys et al., 2014) - including the effect of mountain rain shadows on water yield (Van den Hende et al., 2021). The major reference period for this study was the 1981-1990 period based on the CHIRPs rainfall dataset with data beginning 1981 onwards. We believe the results would be different if an older reference period was used e.g. 100 years ago (presumably actual pristine conditions). This would help to strengthen the findings of this study especially after the evidence of climatic shifts towards wetter conditions in all the water towers. The anthropogenic presence both inside and outside the forested water towers indicates the relevance of local context, and ground research for understanding forest-water-people nexus (Noordwijk et al., 2020)  is recommended.  This will help in understanding in detail the dynamics and co-evolution of coupled human-forest-water systems.*

**Minor edits**

P1,L23: "atmospheric demand" is a bit wonky language for the abstract – could you phrase this differently?

**Answer:** The term "atmospheric demand" in the abstract is now replaced with "potential evapotranspiration"

P1,L35: Consider changing to "Mountain forests capture, store, purify and release water" to avoid ambiguity. Also, was "they" in reference to "mountain forests" or something else?

**Answer:** Revised as suggested to remove the ambiguity. Yes, "They" was in reference to mountain forests.

P2,L40: Are these all the water towers in the region? If so, state that. If not, justify why these towers.

**Answer:** We thank the reviewer for the comment. No, these are not all the water towers in the region. The focus was on elevated forested water towers in the regions (based on humidity scale) sampled in the different East African countries. The definition of the water towers has now been included in the revised manuscript:

*...The selection of water towers was based on aridity index (AI), high elevation, and continuous forest block. The selected water towers have AI≥0.65 (i.e humid), located in high elevated areas under a continuous forest block from the footslope contour to the peak...*

We selected a few of the water towers from different East African countries (see Table 1). For instance, the MAU Forest complex (the largest water tower in the region) did not meet the above criteria as its highest pick is relatively low compared to (Mt Kenya, Aberdare ranges, Mt Elgon, etc). It is also made up of 22 distinct forest blocks with other types of Land uses in between (including urban areas) – which could not meet our pristine assumption. Again, the idea was to sample a few water towers - at least 2 major water towers that met our criteria from each country.

P3,L4: The CRU data set is fairly course and known to contain rather few observations in Africa. Can you justify the use of these data here? Could another remote sensing product provide more accurate data?

**Answer:** We thank the reviewer for the remark. Studies on the East African region suffer from insufficient local-based climate data. Other datasets such as IRI[1] and Maprooms[2] had been considered during the conceptualization of the study, but due to lack of consistency and data gaps among the different countries, were ruled out. The fact that the study looks at the past changes in the elevated forested areas (where there are minimal local measurements upslope of the mountains) warranted going for a dataset that is consistent over different decades and already acceptable in the scientific world.
* * *
[1] https://iri.columbia.edu/resources/enacts/
[2] http://maproom.meteorwanda.gov.rw/maproom/index.html

We also don't see drastic changes in the PET over time (i.e. more of a homogenous pattern) as shown in Fig A3 and pasted below. We argue that the course resolution suffices.

[Figure]

P3,L4: I do not know how CRU gets PET. Could you provide some more information on how these data are prepared? This holds for all the data sets considered.

**Answer:** We thank the reviewer for the comment. More information on data processing of PET, P, and NDVI has been provided in the revised manuscript. The following is a relevant extract from the revised manuscript:

*...Precipitation (P) data were gathered from the Climate Hazards Group Infrared Precipitation with Stations (CHIRPS-v2) with a temporal coverage beginning 1981 and a spatial resolution of 0.05°. CHIRPS uses the Tropical Rainfall Measuring Mission Multi-satellite Precipitation Analysis version 7 (TMPA 3B42 v7) to calibrate global Cold Cloud Duration (CCD) rainfall estimates (Funk et al., 2015). Potential Evapotranspiration (PET) data were sourced from the Climate Research Unit (CRU) database with temporal coverage beginning 1981 and a spatial resolution of 0.5°. The CRU-PET is calculated using the Penman-Monteith formula (Ekström et al., 2007; Harris et al., 2020). Normalized Difference Vegetation Index (NDVI) data to estimate land surface characteristics were sourced from the Global Inventory Monitoring and Modeling System (GIMMS) Third Generation (3 g) Advanced Very High-Resolution Radiometer (AVHRR) sensor onboard the National Oceanic and Atmospheric Administration (NOAA) satellites at a spatial resolution of 0.07° (Kalisa et al., 2019; Pinzon and Tucker, 2014; Tucker et al., 2005) The NDVI is derived using the Bayesian methods with high quality well-calibrated SeaWiFS NDVI data. The resulting NDVI values give an error of ± 0.005 NDVI (Pinzon and Tucker, 2014)...*

P3,L16: Break these longer sections up into sub-section to help the reader follow along.

**Answer:** The sub-sections have now been added in the revised manuscript.

P3,L31: What is "FU"?

**Answer:** This has been revised to Fu which refers to a type of Budyko equation as given by (Zhang et al., 2004).

P4,L11: 2011-2019?

**Answer:** P4,L11 is an equation (4), but we assume you refer to P4,L9. This has been corrected from 201-2019 to 2011-2019

**References**

van der Velde, Y., Vercauteren, N., Jaramillo, F., Dekker, S., Destouni, G., Lyon, S.W. (2014), Exploring hydroclimatic change disparity via the Budyko framework, Hydrological Processes, 28, 4110-4118.

...................................................................................................................

**References included in the responses**

Alcamo, J., Flörke, M. and Märker, M.: Future long-term changes in global water resources driven by socio-economic and climatic changes, Hydrol. Sci. J., 52(2), 247–275, doi:10.1623/hysj.52.2.247, 2007.

Astuti, H. P. and Suryatmojo, H.: Water in the forest: Rain-vegetation interaction to estimate canopy interception in a tropical borneo rainforest, IOP Conf. Ser. Earth Environ. Sci., 361(1), doi:10.1088/1755-1315/361/1/012035, 2019.

Booij, M. J., Schipper, T. C. and Marhaento, H.: Attributing changes in streamflow to land use and climate change for 472 catchments in australia and the United States, Water (Switzerland), 11(5), doi:10.3390/w11051059, 2019.

Bosch, J. M. and Hewlett, J. D.: A review of catchment experiments to determine the effect of vegetation changes on water yield and evapotranspiration, J. Hydrol., 55(1–4), 3–23, doi:10.1016/0022-1694(82)90117-2, 1982.

Budyko, M. I.: Climate and Life., 1974.

Chebet, C.: Environmental degradation to blame for swelling of Rift Valley lakes, Stand. Media, Kenya [online] Available from: https://www.standardmedia.co.ke/environment/article/2001371606/swelling-lakes-of-the-rift-pose-danger-to-residents, 2020.

Chepkoech, A.: Kenya: Rift Valley Lakes Water Levels Rise Dangerously, Dly. Nation, Kenya [online] Available from: https://allafrica.com/stories/202008310228.html (Accessed 15 May 2021), 2020.

Creed, I. and Spargo, A.: Application of the Budyko curve to explore sustainability of water yields from headwater catchments under changing environmental conditions, in Ecological Society of America, August 5-10, 2012. Portland. [online] Available from: http://www.uwo.ca/biology/faculty/creed/PDFs/presentations/APRE47.pdf, 2012.

Creed, I., Spargo, A., Jones, J., Buttle, J., Adams, M., Beall, F. D., Booth, E. G., Campbell, J. L., Clow, D., Elder, K., Green, M. B., Grimm, N. B., Miniat, C., Ramlal, P., Saha, A., Sebestyen, S., Spittlehouse, D., Sterling, S., Williams, M. W., Winkler, R. and Yao, H.: Changing forest water yields in response to climate warming: Results from long-term experimental watershed sites across North America, Glob. Chang. Biol., 20(10), 3191–3208, doi:10.1111/gcb.12615, 2014a.

Creed, I. F., Spargo, A. T., Jones, J. A., Buttle, J. M., Adams, M. B., Beall, F. D., Booth, E. G., Campbell, J. L., Clow, D., Elder, K., Green, M. B., Grimm, N. B., Miniat, C., Ramlal, P., Saha, A., Sebestyen, S., Spittlehouse, D., Sterling, S., Williams, M. W., Winkler, R. and Yao, H.: Changing forest water yields in response to climate warming: Results from long-term experimental watershed sites across North America, Glob. Chang. Biol., 20(10), 3191–3208, doi:10.1111/gcb.12615, 2014b.

Daron, J. D.: Regional Climate Messages for East Africa, Cariaa Assar, 1–30, 2014.

Dey, P. and Mishra, A.: Separating the impacts of climate change and human activities on streamflow: A review of methodologies and critical assumptions, J. Hydrol., 548, 278–290, doi:10.1016/j.jhydrol.2017.03.014, 2017.

Van Dijk, A. I. J. M., Gash, J. H., Van Gorsel, E., Blanken, P. D., Cescatti, A., Emmel, C., Gielen, B., Harman, I. N., Kiely, G., Merbold, L., Montagnani, L., Moors, E., Sottocornola, M., Varlagin, A., Williams, C. A. and Wohlfahrt, G.: Rainfall interception and the coupled surface water and energy balance, Agric. For. Meteorol., 214–215, 402–415, doi:10.1016/j.agrformet.2015.09.006, 2015.

EAC, UNEP and GRID-Arendal: Sustainable Mountain Development in East Africa in a Changing Climate, East African Community, United Nations Environment Programme and GRID-Arendal. Arusha, Nairobi and Arendal. [online] Available from: https://www.grida.no/publications/119, 2016.

Ekström, M., Jones, P. D., Fowler, H. J., Lenderink, G., Buishand, T. A. and Conway, D.: Regional climate model data used within the SWURVE project projected changes in seasonal patterns and estimation of PET, Hydrol. Earth Syst. Sci., 11(3), 1069–1083, doi:10.5194/hess-11-1069-2007, 2007.

Fekete, B. M., Vörösmarty, C. J. and Grabs, W.: High-resolution fields of global runoff combining observed river discharge and simulated water balances, Global Biogeochem. Cycles, 16(3), 15-1-15–10, doi:10.1029/1999gb001254, 2002.

Frank, D. C., Poulter, B., Saurer, M., Esper, J., Huntingford, C., Helle, G. and Treydte, K.: Water-use efficiency and transpiration across European forests during the Anthropocene, , 5(May), doi:10.1038/NCLIMATE2614, 2015.

Funk, C., Peterson, P., Landsfeld, M., Pedreros, D., Verdin, J., Shukla, S., Husak, G., Rowland, J., Harrison, L., Hoell, A. and Michaelsen, J.: The climate hazards infrared precipitation with stations - A new environmental record for monitoring extremes, Sci. Data, 2, 1–21, doi:10.1038/sdata.2015.66, 2015.

Gash, J. H. C., Wright, I. R. and Lloyd, C. R.: Comparative estimates of interception loss from three coniferous forests in Great Britain, J. Hydrol., 48(1–2), 89–105, doi:10.1016/0022-1694(80)90068-2, 1980.

Giannini, A., Lyon, B., Seager, R. and Vigaud, N.: Dynamical and Thermodynamic Elements of Modeled Climate Change at the East African Margin of Convection, Geophys. Res. Lett., 45(2), 992–1000, doi:10.1002/2017GL075486, 2018.

Harris, I., Osborn, T. J., Jones, P. and Lister, D.: Version 4 of the CRU TS monthly high-resolution gridded multivariate climate dataset, Sci. Data, 7(1), 1–18, doi:10.1038/s41597-020-0453-3, 2020.

Helman, D., Lensky, I. M., Yakir, D. and Osem, Y.: Forests growing under dry conditions have higher hydrological resilience to drought than do more humid forests, Glob. Chang. Biol., 23(7), 2801–2817, doi:10.1111/gcb.13551, 2017.

Huntington, T. G.: CO2-induced suppression of transpiration cannot explain increasing runoff, Hydrol. Process., 22(2), 311–314, doi:10.1002/hyp.6925, 2008.

Jiang, C., Xiong, L., Wang, D., Liu, P., Guo, S. and Xu, C. Y.: Separating the impacts of

climate change and human activities on runoff using the Budyko-type equations with time-varying parameters, J. Hydrol., doi:10.1016/j.jhydrol.2014.12.060, 2015.

Kalisa, W., Igbawua, T., Henchiri, M., Ali, S., Zhang, S., Bai, Y. and Zhang, J.: Assessment of climate impact on vegetation dynamics over East Africa from 1982 to 2015, Sci. Rep., 9(1), 1–20, doi:10.1038/s41598-019-53150-0, 2019.

Kiteme, B. P., Liniger, H. and Notter, B.: Dimensions of Global Change in African Mountains : The Example of, , 18–22, 2008.

Knoben, W. J. M., Freer, J. E. and Woods, R. A.: Technical note: Inherent benchmark or not? Comparing Nash-Sutcliffe and Kling-Gupta efficiency scores, Hydrol. Earth Syst. Sci., 23(10), 4323–4331, doi:10.5194/hess-23-4323-2019, 2019.

Krakauer, N. Y., Lakhankar, T. and Anadón, J. D.: Mapping and Attributing Normalized Difference Vegetation Index Trends for Nepal, , 1–15, doi:10.3390/rs9100986, 2017.

Li, C. juan, Chai, Y. qing, Yang, L. sheng and Li, H. rong: Spatio-temporal distribution of flood disasters and analysis of influencing factors in Africa, Nat. Hazards, 82(1), 721–731, doi:10.1007/s11069-016-2181-8, 2016.

Liniger, H., Gikonyo, J., Kiteme, B. and Wiesmann, U.: Assessing and Managing Scarce Tropical Mountain Water Resources, Mt. Res. Dev., 25(2), 163–173, doi:https://doi.org/10.1659/0276-4741(2005)025[0163:AAMSTM]2.0.CO;2., 2005.

Ma, X., Lu, X. X., Van Noordwijk, M., Li, J. T. and Xu, J. C.: Attribution of climate change, vegetation restoration, and engineering measures to the reduction of suspended sediment in the Kejie catchment, southwest China, Hydrol. Earth Syst. Sci., 18(5), 1979–1994, doi:10.5194/hess-18-1979-2014, 2014.

Mamuye, M.: Review on Impacts of Climate Change on Watershed Hydrology, , 8(1), 91–99, 2018.

Marhaento, H., Booij, M. J. and Hoekstra, A. Y.: Attribution of changes in stream flow to land use change and climate change in a mesoscale tropical catchment in Java, Indonesia, Hydrol. Res., 48(4), 1143–1155, doi:10.2166/nh.2016.110, 2017.

Mianabadi, A., Davary, K., Pourreza-Bilondi, M. and Coenders-Gerrits, A. M. J.: Budyko framework; towards non-steady state conditions, J. Hydrol., 588(May), 125089, doi:10.1016/j.jhydrol.2020.125089, 2020.

Niang, I., Ruppel, O. C., Abdrabo, M. A., Essel, A., Lennard, C., Padgham, J. and Urquhart, P.: Africa. In: Climate Change 2014: Impacts, Adaptation, and Vulnerability, in Part B: Regional Aspects. Contribution of Working Group II to the Fifth Assessment Report of the Intergovernmental Panel on Climate Change. [Barros, V.R., C.B. Field, D.J. Dokken, M.D. Mastrandrea, K.J. Mach, T.E. Bilir, M. Chatterjee, K.L. Ebi, Y.O. Estr, edited by V. R. Barros, C. B. Field, D. J. Dokken, M. D. Mastrandrea, and K. J. Mach, pp. 1199–1265, Cambridge University Press, Cambridge., 2014.

Omambia, A. N., Shemsanga, C. and Hernandez, I. A. S.: Climate Change Impacts, Vulnerability, and Adaptation in East Africa (EA) and South America (SA), B. Handb. Clim. Chang. Mitig., 1–4, 573–620, doi:10.1007/978-1-4419-7991-9_17, 2012.

Patel, K.: Rising Waters on Kenya's Great Rift Valley Lakes, Earth Obs. NASA [online] Available from: https://earthobservatory.nasa.gov/images/147226/rising-waters-on-kenyas-great-rift-valley-lakes (Accessed 15 May 2021), 2020.

Pinzon, J. E. and Tucker, C. J.: A non-stationary 1981-2012 AVHRR NDVI3g time series, Remote Sens., 6(8), 6929–6960, doi:10.3390/rs6086929, 2014.

Røhr, P. C. and Killingtveit, Å.: Rainfall distribution on the slopes of Mt Kilimanjaro, Hydrol. Sci. J., 48(1), 65–77, doi:10.1623/hysj.48.1.65.43483, 2003.

Schaake, J. S.: From climate to flow., 1990.

Sinha, J., Sharma, A., Khan, M. and Goyal, M. K.: Assessment of the impacts of climatic variability and anthropogenic stress on hydrologic resilience to warming shifts in Peninsular India, Sci. Rep., 8(1), 1–14, doi:10.1038/s41598-018-32091-0, 2018.

Tech, J.: About SWAT+ - SWAT+ Documentation. [online] Available from: https://swatplus.gitbook.io/docs/, 2019.

Teng, J., Chiew, F. H. S., Vaze, J., Marvanek, S. and Kirono, D. G. C.: Estimation of climate change impact on mean annual runoff across continental Australia using Budyko and Fu equations and hydrological models, J. Hydrometeorol., 13(3), 1094–1106, doi:10.1175/JHM-D-11-097.1, 2012.

Teuling, A. J.: A Forest Evapotranspiration Paradox Investigated Using Lysimeter Data, Vadose Zo. J., 17(1), 170031, doi:10.2136/vzj2017.01.0031, 2018.

Teuling, A. J., De Badts, E. A. G., Jansen, F. A., Fuchs, R., Buitink, J., Van Dijke, A. J. H. and Sterling, S. M.: Climate change, reforestation/afforestation, and urbanization impacts on evapotranspiration and streamflow in Europe, Hydrol. Earth Syst. Sci., 23(9), 3631–3652, doi:10.5194/hess-23-3631-2019, 2019.

Tucker, C. J., Pinzon, J. E., Brown, M. E., Slayback, A., Pak, E. W., Mahoney, R., Vermote, E. F. and Saleous, N. E. L.: An extended AVHRR 8-kni NDVI dataset compatible with MODISand SPOT vegetation NDVI data, Int. J. Remote Sens., 26(20), 2005.

Ulrich, A., Ifejika Speranza, C., Roden, P., Kiteme, B., Wiesmann, U. and Nüsser, M.: Small-scale farming in semi-arid areas: Livelihood dynamics between 1997 and 2010 in Laikipia, Kenya, J. Rural Stud., doi:10.1016/j.jrurstud.2012.02.003, 2012.

Van der Velde, Y., Vercauteren, N., Jaramillo, F., Dekker, S. C., Destouni, G. and Lyon, S. W.: Exploring hydroclimatic change disparity via the Budyko framework, Hydrol. Process., 28(13), 4110–4118, doi:10.1002/hyp.9949, 2014.

Wambua, C.: Why Kenya's Rift Valley lakes are going through a crisis, Aljazeera [online] Available from: https://www.aljazeera.com/news/2020/08/30/why-kenyas-rift-valley-lakes-are-going-through-a-crisis/, 2020.

Wei, X. and Zhang, M.: Research Methods for Assessing the Impacts of Forest Disturbance on Hydrology at Large-scale Watersheds, Landsc. Ecol. For. Manag. Conserv., (May), 119–147, doi:10.1007/978-3-642-12754-0_6, 2011.

Yuan, W., Piao, S., Qin, D., Dong, W., Xia, J., Lin, H. and Chen, M.: Influence of Vegetation Growth on the Enhanced Seasonality of Atmospheric CO2, Global Biogeochem. Cycles, 32, 32–41, doi:https://doi.org/10.1002/ 2017GB005802, 2017.

Zeng, F., Ma, M. G., Di, D. R. and Shi, W. Y.: Separating the impacts of climate change and human activities on runoff: A review of method and application, Water (Switzerland), 12(8), 1–17, doi:10.3390/W12082201, 2020.

Zhang, L., Hickel, K., Dawes, W. R., Chiew, F. H. S., Western, A. W. and Briggs, P. R.: A rational function approach for estimating mean annual evapotranspiration, Water Resour. Res., 40(2), 1–14, doi:10.1029/2003WR002710, 2004.

Zhang, M., Wei, X., Sun, P. and Liu, S.: The effect of forest harvesting and climatic variability on runoff in a large watershed: The case study in the Upper Minjiang River of Yangtze River basin, J. Hydrol., 464–465, 1–11, doi:10.1016/j.jhydrol.2012.05.050, 2012.

Zimmermann, L., Frühauf, C. and Bernhofer, C.: The role of interception in the water budget of spruce stands in the Eastern Ore Mountains/Germany, Phys. Chem. Earth, Part B Hydrol. Ocean. Atmos., 24(7), 809–812, doi:10.1016/S1464-1909(99)00085-4, 1999.

---

## Author Comment (AC2)

**RC1**: **'Comment on hess-2021-151'**, M. van Noordwijk

General

1. The manuscript provides an interesting comparative study of the 'water towers' in East Africa and the change in terms of a simple water balance that can be inferred from a combination of various spatial data sources

We thank the reviewer for the constructive comments. We have considered all comments and suggestions in our revised manuscript.

2. The description of the quantitative framework can be improved, including a more consistent use of acronyms (especially for actual evapotranspiration) and equations

**Answer:** We thank the reviewer for the feedback. The use of acronyms has been revised to ensure consistency. In the revised Manuscript, the acronym for actual evapotranspiration is ET, potential evapotranspiration is PET, and Water yield is Q.

3. The study relies heavily on the use of a link between NDVI and the omega parameter in the Budyko framework, while the text acknowledges many factors (including soil, topography and seasonality) influence the relationship. At least in the discussion this needs some further work to see how much this could have influenced results and conclusions.

**Answer:** We thank the reviewer for this comment. We have revised the section that acknowledges many factors such as soil types, topography, seasonality and how they are connected to vegetation signatures in our study and hence our confidence of adopting NDVI-based formulation. We have also incorporated a discussion of uncertainties/limitations in the discussion section. The following are relevant extracts from the revised manuscript:

In the methodology section we have revised the text as follows:

*...The $\omega$ parameter is the most difficult parameter to estimate in Budyko framework applications (Bai et al., 2019). It reflects the impact of other factors such as land surface characteristics and climate seasonality on water and energy balances (Li et al., 2013). Previous studies have adopted various ways to estimate the $\omega$ parameter. Some studies used fitted values based on land-use/cover of the areas under investigation. For instance, Zhang et al. (2012) used values of 2, 0.5, and 1 to represent $\omega$ for the forest, grassland and shrubland respectively. Creed et al. (2014) used $\omega = 2$ in forested catchments, $\omega = 0.5$ in grassland or cropland catchments, and $\omega = 1$ in mixed cover catchments. Other studies calibrated it based on historical data (Gunkel and Lange, 2017; Redhead et al., 2016; Yang et al., 2014). However, for data-limited regions, calibration-based estimations are impossible and simpler methods to estimate the $\omega$ parameter based on readily available data are desirable. Land surface hydrology varies due to variations in different factors such as vegetation, soil types, topography, and climate seasonality (Li et al., 2013; Yan et al., 2020). Soil texture and topography influence the amount of water available for vegetation – hence the vegetation signatures can reflect the underlying conditions of soil water conditions, topography, seasonality etc. Donohue et al. (2007) argued based on the theory of ecohydrological equilibrium that in water-limited environments, vegetation is the*

*integrated response to all processes affecting the availability of water. Therefore, vegetation information can serve as a good integrated indicator of these ecohydrological impacts on water and energy balances as it reflects the integrated landscape and climatic features. Using data from 26 major global river basins under a wide range of climate regimes, Li et al. (2013) developed a simple parameterization for Budyko ω parameter based solely on vegetation information as shown in equation 2 (in pre-print version)...*

When discussing possible sources of uncertainties in the discussion section, the following text has been included:

*...Uncertainties may also arise in the general assumption that estimation of land surface characteristics (ω) based on NDVI formulation provides values that represent integrated conditions for soil, topography, and climate seasonality. Some studies using various hydrological approaches have reported the significance of these factors in influencing catchment hydrology (Kirkby et al., 2002; Troch et al., 2013; Western et al., 2004; Woods, 2002). There is a need for more research to come up with methodological consistency in estimating ω parameters when using the Budyko framework...*

4. The eight water towers are most described as 'replicates', rather than each having a specific geographic, ecological and social context: this may be the limit of what is currently possible, but at least some of the contrasts noted call for further analysis and attribution (e.g. in relation to human population density within and surrounding the water tower.

**Answer:** We completely agree that the eight water towers should not be seen as 'replicates'. We have added Table A2 on 'description of the selected water towers' that gives brief information on geographic, ecological, social context etc of each water tower in the revised manuscript. We have also acknowledged the relevance of local context and the need for ground research for understanding the forest-water-people nexus in the discussion section. See the following relevant extract from the revised manuscript:

*...The anthropogenic presence both inside and outside the forested water towers indicates the relevance of local context, and ground research for understanding forest-water-people nexus (Noordwijk et al., 2020) is recommended. This will help in understanding in detail the dynamics and co-evolution of coupled human-forest-water systems...*

5. It would help the paper if sharper questions would be formulated at the end of the introduction that gives structure to the subsequent discussion.

**Answer:** We thank the reviewer for the remark. We have now constructed the research question as follows:

"What are the effects of climate and land-use changes on water yield for the selected forested water towers?"

We hypothesized that, in areas considered as pristine or protected zones (i.e. high elevated forested areas), with AI≥0.65, changes in water yield would majorly be attributed to climate changes and negligibly due to land use/cover changes. The high elevated forested areas would then be expected to fall on the reference Budyko curve over the study period.

6. Beyond the supply of blue water to downstream parts of the watershed, the high actual evapotranspiration in water towers plays a role in regional rainfall recycling -- at least some discussion of this aspect would be relevant.

**Answer:** We thank the reviewer for bringing this relevant issue to our attention. We have included this point in our revised discussion, i.e. the importance of high elevated forested areas in maintaining high actual evapotranspiration – a key component in regional rainfall recycling. The following is the relevant extract from the revised manuscript:

*...Our study offers important findings on the sensitivity of water yield to climate and land-use changes and the importance of these water towers in the generation and supply of blue water to adjacent lowland areas. These results can be used by decision-makers, policymakers, stakeholders, and scientists to emphasize the need to protect and conserve the high elevated forested areas in the region, particularly forest ecosystems above 2000 m a.s.l – where there is a surplus of blue water. The Budyko framework provides primary results that can inform detail hydrological assessments. For instance, our findings show that elevated forested water towers are important areas for maintaining high ET in the region. This finding can be explored further by studying the role of water towers in the supply of green water in the region (i.e. the role of water towers in regional rainfall/moisture recycling) (Ellison et al., 2017; Keys et al., 2014) - including the effect of mountain rain shadows on water yield (Van den Hende et al., 2021)...*

**Minor**

p1, Line 17 Mention 'steady state' assumption of Budyko framework at an annual time scale

**Answer:** This has been corrected in the revised manuscript.

p1, Line 24 'non-resilient' suggests a binary classification, is there a more gradual description on the degree of resilience

**Answer:** We thank the reviewer for this remark. The resilience/non-resilience of the water towers is based on Budyko metrics elasticity (e) values as discussed in (Creed et al., 2014; Helman et al., 2017; Sinha et al., 2018) and calculated as a ratio of DI ranges to EI ranges (as shown in Eq 6). For simplicity, the terms low and high elasticity values were used in our study to represent the minimum empirical e value and maximum e value respectively for the different water towers. Elastic catchments (i.e. high elasticity) are expected to plot along the Budyko curve (i.e. resilient to climate changes) while inelastic catchments (i.e. low elasticity) (non-resilience to climate changes) would deviate from the Budyko curve.

p1, Line 29 but mountains also cause 'rainshadows' that don't get the rainfall they might have had without the presence of a mountain...

**Answer:** We thank the reviewer for this comment. In the revised manuscript we have recognized the effect as an area that requires further investigations. See the relevant extract below:

*...This finding can be explored further by studying the role of water towers in the supply of green water in the region (i.e. the role of water towers in regional rainfall/moisture recycling) - including the effect of mountain rain shadows on water yield...*

p1, Line 31 more quantitative criteria are needed to get the type of delineation that you use here

**Answer:** We thank the reviewer for this comment. More information regarding the criteria we used in definition of water towers has been added in the introduction section. The following is the relevant extract from the revised manuscript:

*...The selection of water towers was based on aridity index (AI), high elevation, and continuous forest block. The selected water towers have AI≥0.65 (i.e humid), located in high elevated areas under a continuous forest block from the footslope contour to the peak.*

*The montane forests are the three major forest ecosystems defined and delineated by (EAC et al., 2016; UNEP, 2010), and they include the Albertine Rift, the Kenyan Highlands, and the Ethiopian Highlands. They were defined and delineated based on major rivers in the region. All the selected water towers in this study fall in the three forest ecosystems....*

p1, Line 34 in glaciated mountain chains water flow depends primarily on temperature, without ice cap on recent rainfall -- so the temporal variability will differ and dependence on land cover increase

**Answer:** We thank the reviewer for this remark. This important point *'temperature is a key factor in determining water flows from glaciated mountain chains'* has been included in our revised manuscript.

p1, Line 35 'receive' is a rather passive description -- isn't it 'convert atmospheric moisture into rainfall'
**Answer:** We thank the reviewer for the remark. This has been revised to *'maintain significantly more precipitation than adjacent lowlands....'*

The intention is to introduce a reasoning that elevated forested areas record higher rainfall compared to lower slopes – an imporatnt source of water resources, especially for drier lowlands.

p1, Line 37 Some reference to Africa as geologically old shield, but rift valley plate tectonics are associated with younger and higher mountains

**Answer:** We thank the reviewer for the remark. This has been added in the revised manuscript.

*...It is also paramount to mention that the East-African rift system has extensive plate tectonics that are considered relatively recent (Dawson, 2008)...*

p1, Line 39-40 If you introduce more quantitative P/Epot criteria in line 31, this discussion on E African water towers becomes more meaningful, as it relates to both the P and the Epot side of the ratio.

**Answer:** We thank the reviewer for this comment. We have provided a clear definition of the water towers including quantitative P/Epot criteria.

*...The selected water towers have AI≥0.65 (i.e humid), located in high elevated areas under a continuous forest block from the footslope contour to the highest point...*

p1, Line 41 rainfall distribution is meager? what do you mean

**Answer:** We thank the reviewer for this comment. The intention is to describe that there is a high dependency on rainfall in the East African region, but rainfall distribution is insufficient/scanty. We have replaced the word 'meager' with 'insufficient'

p1, Line 42 Early work on rainfall in Sudan (El Tom, 1972) showed that the standard deviation of annual rainfall is nearly independent of mean annual value, showing that dry areas are highly variable in relative terms, with decadal variation super-imposed (Hulme, 1990) and not easily distinguishable from trended global climate change.

**Answer:** We thank the reviewer for the remark. We have expanded the text by acknowledging the information from these early works of El Tom, 1972 and Hulme 1990.

*...El Tom (1972) tested the reliability of rainfall and showed that in the dry areas, the rainfall is highly variable and nearly independent of the mean annual value – affecting rainfed agriculture in the region. Fluctuations of rainfall are evident in both seasonal and decadal time series mainly in the semi-arid zones (Hulme, 1990)...*

p2, Line 5 Please unpack the sentence
**Answer:** We thank the reviewer for this comment. We have revised the sentence as shown in the following extract:

*...Understanding historical climate and human-induced land-use changes and their impacts on streamflow can explain some of the hydrological events experienced in the adjacent lowlands. This can help inform the role of forested water towers in observed extremities in the lowlands such as floods and hydrological droughts...*

p2. Line 6 Possibly relevant: ET estimates for SS Africa

**Answer:** We thank the reviewer for this comment. However, we suspect SS Africa refers to Sub-Saharan Africa, but the comment is not clear to us when we connect it to Page 2, line 6. We had sent an email seeking clarification but had not received a reply by the time of posting our responses. Nevertheless, based on the preceding comment, we have unpacked the sentence as shown in the above response.

p2, Line 8 For corrections on common deforestation discourses, see Aleman et al. 2018
**Answer:** We thank the reviewer for sharing this important reference, we reviewed the publication added the reference in the revised manuscript. See the following extract from the revised manuscript:

*...To our knowledge, there are no studies that have focused on the East-African forested water towers and their ability to generate streamflow under a changing climate and land-use in the East-African region. At the regional scale, studies in the region either focus on studying forest trends/deforestation (Aleman et al., 2018) or the effects of land-use changes on climate (Otieno and Anyah, 2012). At the river basin scale, studies in the region focus on hydrological responses (Gabiri et al., 2020; Hyandye et al., 2018; Mango et al., 2011)...*

p2, Line 16 The methods of Ma et al. 2010, 2014 combine these two categories by running rainfall statistics and recorded land-use change patterns in reverse order in calibrated process-based models

**Answer:** We thank the reviewer for the remark. The reference has been acknowledged in the revised manuscript. The following is the relevant extract from the revised manuscript:

*...Various approaches have been used for studying the effects of climate and land-use changes on streamflow. Jiang et al. (2015) categorized such methods into two: (a) deterministic rainfall-runoff models and (b) statistical methods. Ma et al. (2014) combined the two categories by running rainfall statistics and recorded land-use change patterns in reverse order in calibrated process-based models. Dey and Mishra (2017) reviewed the existing approaches and categorized these approaches into four categories; (i) experimental approach e.g. paired catchment method...*

P2 Line 22 Maybe mention the steady-state assumptions at annual time-scale upfront. A simple equation might help here.

**Answer:** We thank the reviewer for the remark. This has been revised accordingly in the revised manuscript. A simple water balance equation has also been included as P=ET+Q under steady-state conditions (as Equation 1 in the revised manuscript):

*...A steady-state is reached when the total input (i.e., precipitation) equals the total output (i.e., evapotranspiration and water yield) (Han et al., 2020) and changes in soil water storage are zero (Donohue et al., 2007). Hence a simple water balance equation assuming steady-state conditions can be written as:*

$$P = ET + Q \tag{1}$$

*where P is precipitation, ET is actual evapotranspiration and Q is water yield...*

p2 Line 26 It would help the subsequent discussion if you formulate some clear questions here that you try to answer in the results section

**Answer:** We thank the reviewer for this comment. We have now constructed the research question as follows:

"What are the effects of climate and land-use changes on water yield for the selected forested water towers?"

We hypothesized that, in areas considered as pristine or protected zones (i.e. high elevated forested areas), with AI≥0.65, changes in water yield would majorly be attributed to climate changes and negligibly due to land use/cover changes. The high elevated forested areas would then be expected to fall on the reference Budyko curve over the study period.

P2 line 30 if you want to avoid use of 'we', please find a less abstract passive formulation...

**Answer:** We thank the reviewer for the remark. This has been revised accordingly and adopted the use of 'we'.

Fig. 1 As 'montane forests' and 'water towers' only partially overlap, please give the quantitative definitions of both;

**Answer:** We thank the reviewer for this comment. The definition of the water towers is now included in the revised manuscript as:

*...The selection of water towers was based on aridity index (AI), high elevation, and continuous forest block. The selected water towers have AI≥0.65 (i.e humid), located in high elevated areas under a continuous forest block from the footslope contour to the peak...*

The definition of the montane forests is given as:

*...The montane forests are the three major forest ecosystems defined and delineated by (EAC et al., 2016; UNEP, 2010), and they include the Albertine Rift, the Kenyan Highlands, and the Ethiopian Highlands. They were defined and delineated based on major rivers in the region. All the selected water towers in this study fall in the three forest ecosystems...*

Public discussion on the Mau forests in Kenya described these as 'water towers', you don't; again clarifying the quantitative criteria can help

**Answer:** We thank the reviewer for this comment. You are right, Mau Water Tower is the largest 'water tower' in East Africa (EAC et al., 2016; Odawa and Seo, 2019). We selected a few of the water towers based on elevated forest areas (highest land areas) under one continuous forest block. MAU Forest complex did not meet our criteria as its highest pick is relatively low compared to (Mt Kenya, Aberdare ranges, Mt Elgon etc). It is also made up 22 distinct forest blocks with other types of Land uses in between (including urban areas) – which could not meet our pristine assumption. Again, the idea was to sample a few water towers from each country. In Kenya, 2 water towers were considered sufficient (Mt Kenya, Aberdare ranges and part of Mt Elgon).

Table 1 In the Dewi et al. water tower delineation no fixed contour was used for the delineation, but one relative to the watershed as a whole. Please clarify your choice here, esp regarding the two (Aberdare and Bale) that were adjusted to the surrounding areas...

**Answer:** We thank the reviewer for this comment. As explained in the above response, the choice of contour delineation was expected to give us a continuous forest block from the selected footslope contour to the highest peak. The adjustment was done upwards and not downwards. Bale mountains rises from 2,500 m a.s.l. (Hillman, 1988). We adjusted upwards to 2,600 m a.s.l to ensure we capture majorly the elevated forested areas presumably under pristine conditions. We could not settle for the administrative boundary at 2,500 m a.s.l as it was capturing more of the surrounding areas. A similar approach was also applied for Aberdare ranges as shown below:

[Figure]

**Bale mountains:** When the official forest boundary was selected (i.e. 2,500 m a.s.l - cyan blue contour), large surrounding areas were captured – an adjustment to 2,600 m a.s.l (black contour) gave a continuous forest block up to the peak of 4,337 a.s.l.

**Aberdare ranges**: Similarly, when the 2,000 m contour was selected (cyan blue contour), large areas were captured, hence an upward adjustment to 2,100 m a.s.l (black contour)

p3 Line 11 Here you seem to shift from PET to ET or ETa -- the preceding paragraph only mentions Epot.

**Answer.** We thank the reviewer for this comment. The potential evapotranspiration mentioned in the preceding is as part of global datasets used in the study. The PET together with P and NDVI datasets were used to calculate actual evapotranspiration (ET) using the Budyko equation (i.e. we used PET to calculate ET).

p3 Line 14 This section may be clearer if you first present a water balance equation...
**Answer:** We thank the reviewer for this comment. A simple water balance equation has been included as P=ET+Q under steady-state conditions.

p3 Line 17 Deserts tend to have wadi's -- even in zones with low average rainfall, runoff occurs and rainfall intensity exceeds instantaneous infiltration capacity. Your Budyko-based description here needs some empirical adjustment (and scale considerations)

**Answer:** We thank the reviewer for the remark. In the discussion section and especially when discussing the potential sources of uncertainties, we have acknowledged the lack of empirical adjustment of the Budyko framework to capture special features in the lowland areas such as desert wardis:

*...Although the focus of the study was in the elevated forested areas, empirical adjustment of the Budyko model may be needed to capture special features such as desert wadis in the application of the Budyko equation in the lowland areas...*

p3 Line 17 Not only under very dry conditions... About 50% of tropics has a P/PET ratio below 0.65; only a quarter has P/PET above 1.0

**Answer:** We thank the reviewer for this comment. We have added a reference and related information in our revised manuscript:

*...About 60% of the world's land surface is considered an arid area (i.e. P/PET ratio also know as aridity index (AI) of below 0.65 (Convention on Biological Diversity, 2001)...*

p3 line 31 Fu in stead of FU

**Answer:** This has been corrected.

p3 Equation 1 -- please specify the time step (1 year?)
**Answer:** We thank the reviewer for this comment. The time step has been included. The time step is 10 years between 1981 and 2010 and 9 years between 2011 and 2019.

Wouldn't it be better to include a DeltaS storage term, and then make explicit that you assume this is zero at the time scale of your analysis (but this is a considerable source of uncertainty and error...)
**Answer:** We thank the reviewer for this comment. This has been revised. The $\Delta S$ has been included in the equation.

Fig 1A please settle on a single acronym for AET = ETa = ET
**Answer:** We thank the reviewer for this comment. This has been corrected: Actual evapotranspiration now represented with an acronym ET

p3 Line 37 The seasonality effect is linked to the DeltaS term that you're hiding...

**Answer:** We thank the reviewer for this comment. The $\Delta S$ is now acknowledged and shown in the revised equation:

$$Q = P - ET + \Delta S$$

p4 line 6 As this is an empirical result, please describe the data set on which it was calibrated (from which it was derived)

**Answer:** We thank the reviewer for this comment. We have now mentioned the kind of datasets used in the revised manuscript.

*...Li et al. (2013) used data from 26 major global river basins under a wide range of climate regimes to come up with a simple parameterization for Budyko $\omega$ parameter based solely on vegetation information...*

p4 Line 8 So what about other influences on omega (soil types, and topography, climate seasonality, ...) that you just mentioned? You assume that these are at the average values in the Li et al. dataset? This will require some further justification, especially as you operate in the relatively rare bimodal rainfall part of the world.

**Answer:** We thank the reviewer for this comment. We have revised the text on estimations and key assumptions around Budyko $\omega$ parameter in our study. The following is the relevant extract from the revised manuscript:

*...The $\omega$ parameter is the most difficult parameter to estimate in Budyko framework applications (Bai et al., 2019). It reflects the impact of other factors such as land surface characteristics and climate seasonality on water and energy balances (Li et al., 2013). Previous studies have adopted various ways to estimate the $\omega$ parameter. Some studies used fitted values based on land-use/cover of the areas under investigation. For instance, Zhang et al. (2012) used values of 2, 0.5, and 1 to represent $\omega$ for the forest, grassland and shrubland respectively. Creed et al. (2014) used $\omega = 2$ in forested catchments, $\omega = 0.5$ in grassland or cropland catchments, and $\omega = 1$ in mixed cover catchments. Other studies calibrated it based on historical data (Gunkel and Lange, 2017; Redhead et al., 2016; Yang et al., 2014). However, for data-limited regions, calibration-based estimations are impossible and simpler methods to estimate the $\omega$ parameter based on readily available data are desirable. Land surface hydrology varies due to variations in different factors such as vegetation, soil types, topography, and climate seasonality (Li et al., 2013; Yan et al., 2020). Soil texture and topography influence the amount of water available for vegetation – hence the vegetation signatures can reflect the underlying conditions of soil water conditions, topography, seasonality etc. Donohue et al. (2007) argued based on the theory of ecohydrological equilibrium that in water-limited environments, vegetation is the integrated response to all processes affecting the availability of water. Therefore, vegetation information can serve as **a good integrated indicator** of these ecohydrological impacts on water and energy balances as it reflects the integrated landscape and climatic features. Using data from 26 major global river basins under a wide range of climate regimes, Li et al. (2013) developed a simple parameterization for Budyko $\omega$ parameter based solely on vegetation information as shown in equation 2 (preprint version)...*

p4 Line 17 Please remove ;
**Answer:** This has been removed.

p4 line 12 Please indicate what you treat as 'known' inputs here and what as parameters to be estimated
**Answer:** We thank the reviewer for this comment. This has been revised accordingly. Known inputs are P (from datasets) and ET (Estimated through the Budyko model). The parameter to be estimated here is Q

p4 Line 36 So the EIBud is based on the NDVI relationship? It would help if you give more formal definitions of the terms here

**Answer:** We thank the reviewer for this comment. In this study, the Budyko curve was developed based on 1981-1990 conditions (for each of the water tower). This was assumed to represent the reference condition for the water balance, to effectively assess the trends in the succeeding periods of 1991-2000, 2001-2010, and 2011-2019.

We have added the following information to give more clarity:
*...The $EI_{Bud}$ therefore represents the 'theoretical value' (i.e. point on the Budyko curve where the water tower was expected to fall at a particular period. The $EI_{Sim}$ represents the point where the water tower plotted in that period. The difference between the expected/reference point ($EI_{Bud}$) and the actual point ($EI_{Sim}$) was then calculated to give the deviation (d) from the Budyko curve...*

p4 Line 6 Is the DeltaEI here the same as d in Eq 5?

**Answer:** We thank the reviewer for this comment. The $\Delta EI$ in Equation 6 is not exactly the same as the deviation ($d$) in equation 5. We have added the following text to ensure there is clarity:

*...Elasticity (e) was defined as the ratio of interdecadal variation in dryness index (DI) to interdecadal variation in the evaporative index (EI)...*

*...In our study, the evaporative indices (EI) for the four periods (i.e. 1981-1990, 1991-2000, 2001-2010, and 2011-2019) were calculated (based on averages of ET and P for each period). The range in EI (i.e. $\Delta EI$) is the difference between the reference/baseline EI (of 1981-1990) and succeeding periods of 1991-2000, 2001-2010, and 2011-2019...*

For instance:
i. EI of 1991-2000 minus reference EI of 1981-1990 gives the 1$^{st}$ range $\Delta EI_A$
ii. EI of 2001-2010 minus reference EI of 1981-1990 gives the 2$^{nd}$ range $\Delta EI_B$
ii. EI of 2010-2019 minus reference EI of 1981-1990 gives the 3$^{rd}$ range $\Delta EI_C$

The $d$ in equation 5 is basically the vertical deviation from the point at which the water tower was expected to fall ($EI_{Bud}$) i.e. reference point (i.e. determined by the Budyko curve) versus the actual point that was simulated ($EI_{Sim}$).

p5 Wouldn't it be easier and more informative to present the ETa/PET ratios?

**Answer:** We thank the reviewer for this comment. Yes, we agree that the ratio of ET/PET can be very informative especially if the focus is on water stress and land-atmosphere interaction rather than water yield. In the end, our goal is water yield (Q), and Et/PET does not provide much additional insight. The ratio is linked to canopy–atmosphere coupling and is very informative as it can characterize how different areas contribute to the 'production of 'green water' – a key component in moisture/rainfall recycling. This ratio is commonly applied together with the ratio of Q/P or P/PET in Turc space illustrations – an alternative framework to using the Budyko framework.

The ratio of ET/PET has been used differently by various studies. For instance, to estimate ET in hydrological models, to estimate irrigation requirements, to monitor crop water stress etc as described in (Peng et al., 2019).

We feel that the ET/PET ratios can be very informative for studies focusing on 'green water', regional rainfall recycling, and soil moisture conditions. Our study focused on the generation and supply of 'blue water' from high elevated forested areas. We feel more comfortable using Fu's equation in our study as the focus is on the sensitivity of 'blue water' from high elevated forested zones. We have, however, acknowledged the 'green water' as the study already shows that water towers are important areas that maintain high ET and further studies on regional rainfall recycling can further improve our results on the importance of water towers in the East African region. See the relevant extract below:

*...our findings show that elevated forested water towers are important areas for maintaining high ET in the region. This finding can be explored further by studying the role of water towers in the supply of green water in the region (i.e. the role of water towers in regional rainfall/moisture recycling) - including the effect of mountain rain shadows on water yield...*

p7 line 1 where omega values 'observed'? maybe 'derived'

**Answer:** We thank the reviewer for this comment. This has been noted and replaced 'observed' with 'derived'

Fig 7 How can Q estimates of 1000 mm/year be obtained for places with P hardly above 1000 mm/year?

**Answer:** We thank the reviewer for this comment. The Q estimates of 1000 mm/year were mainly observed in the western part where a Rainfall of over 2000 mm/year was observed as shown in Fig 2A - also in some parts within the Mt Kilimanjaro where annual Rainfall was of over 2000 mm/year. Our study observed an over-estimation of simulated Q when a comparison with GRDC runoff was done. This is now highlighted in the discussion of uncertainties. The following is the relevant extract from the revised manuscript.

*...In this study, the spatial pattern of the simulated streamflow in the Budyko framework closely resembles the pattern observed in the GRDC composite runoff. We however noted overestimation of water yield in the comparison. This type of observation was also reported by (Teng et al., 2012), where the Budyko equation was found to overestimate water yield in drier regions. Moreover, other factors such as soil type, topography, seasonality, water storage, interception, etc were not accounted for in the quantitative framework which can affect the simulations in the selected forested water towers.*

*Canopy interception, for instance, plays an important role in the water balance of forested ecosystems as noted in several studies (Astuti and Suryatmojo, 2019; Gash et al., 1980; Teuling et al., 2019; Zimmermann et al., 1999). In their study, (Teuling et al., 2019) found many forested points to have average yearly evapotranspiration (ET) that exceeds the average potential evapotranspiration (PET). Van Dijk et al. (2015) opined that this is possible due to underestimation of evapotranspiration which was attributed to evaporation of interception water by energy not captured in the formulation of PET. The forest evapotranspiration paradox is further discussed in (Teuling, 2018). The correction of underestimation in (Teuling et al., 2019) indicates the need for long-term lysimeter observations for studies focussing on forested ecosystems. Availability of meteorological data in the upper slopes of the East African mountains is a big gap as the majority of meteorological observations are conducted below 1500 m a.s.l and most of the upper slopes data rely on extrapolation of hydrological analysis in the lowlands (Røhr and Killingtveit, 2003).*

*Local-based runoff measurements would have helped to interpret if there is indeed an overestimation in our study. That said, we observed positive KGE which indicates a "good" model performance...*

p14 Discussion: A clearer structure of the discussion is needed.

**Answer:** We thank the reviewer for the comments in the discussion section. The following comments together with related comments from the other referees helped in improving the discussion section.

p14 Line 19 As you used NDVI data, you used land cover rather than land use change as basis...

**Answer:** We thank the reviewer for this comment. A discussion on uncertainties has now been included. The following is the relevant extract from the revised manuscript:

*...Uncertainties may also arise in the general assumption that estimation of land surface characteristics ($\omega$) based on NDVI formulation provides values that represent integrated conditions for soil, topography, and climate seasonality. Some studies using various hydrological approaches have reported the significance of these factors in influencing catchment hydrology (Kirkby et al., 2002; Troch et al., 2013; Western et al., 2004; Woods, 2002). There is a need for more research to come up with methodological consistency in estimating $\omega$ parameters when using the Budyko framework...*

p14 Line 21 The sensitivity to land cover change reflects the limited degree of actual change (due to existing institutional arrangements) rather than the lack of response if such rules would be relaxed. Please distinguish these two aspects.

**Answer:** We thank the reviewer for this remark. We have acknowledged the existing institutional arrangements in our discussion. The following is a relevant extract:

*...Our results indicate that changes in precipitation and potential evapotranspiration are the major determinants of blue water availability from high elevated forested water towers in the East African region. However, it would also be paramount to point out that lack of evidence of sensitivity to land-use changes within the water towers, may also be due to existing institutional arrangements, hence a limited degree of actual change. We presume that the results would be different if such rules would be relaxed...*

p14 line 24. An alternative to describing deviations along the Y axis (vertical) is to attribute them along the X-axis (horizontal): would such an approach be feasible?
**Answer:** We thank the reviewer for this comment. Yes, this is possible. Actually, we had discussed the horizontal shifts in the pre-print version (p15 line 6-14). The following is the relevant extract from the revised manuscript:

*...Further illustrations can be shown in the Budyko space based on the horizontal shifts relative to the dryness index (DI). The horizontal shifts are important indicators of the behavior of the water towers towards warmer or humid conditions. These horizontal deviations reflect a change in the climatic conditions specifically, temperature and precipitation (Creed and Spargo, 2012). This study observed that the majority of the water towers (7 out of 9) plotted within humid conditions (i.e. DI <1). On the other hand, two of the water towers (i.e. Mt Meru and Bale mountains) demonstrated warmer conditions (i.e. DI >1). One major observation is that water towers in Eastern Africa seem to shift towards the left, an indication of the increased humid conditions especially in the period of 2011-2019. At the same time, a gradual increase in PET was observed in all the water towers. A climate shift to wetter conditions and simultaneous increases in regional temperatures have also been reported in the East African region and projected to increase by the end of 21st century (Giannini et al., 2018; Niang et al., 2014; Omambia et al., 2012)...*

p14 Discussion: Can you imagine doing the same analysis on the basis of ET/PET ratios attributed to NDVI, rather than the more complex Budyko route that involves P in the estimation of omega?

**Answer:** We thank the reviewer for the remark. However, we do not see how P can be removed from the estimation of ET. If one would investigate this system only with NDVI as a predictor of ET, then you would miss important changes due to P.

p14 Line 27: what do you mean by naturally occurring oscillations in this context? Does the occurrence of fire (partially anthropogenic) play a role: it changes NDVI for one or more years, increasing water yield; it may be more common on e.g. Mt Kenya and in the Imatong mountains

**Answer:** We thank the reviewer for this remark. We have removed the term 'naturally occurring oscillations in the revised manuscript. The following is the relevant extract from the revised manuscript:

*...The two water towers where no deviations were observed (i.e. Mt Elgon and Imatong mountains) indicate that the hydroclimatic conditions in the study period did not vary much from the reference conditions of 1981-1990 and any changes in water yield in the two water towers can largely be associated with climatic changes in P and PET...*

p15 Line 4 Please clarify 'resilience' as bouncing back in relation to 'elasticity' that refers to the degree of initial change, rather than its temporal dimension.

**Answer:** We thank the reviewer for this comment. Actually, in our calculation of elasticity (see equation 6), all the DI and EI ranges are referenced to the starting period of 1981-1990 to determine the degree of change from initial conditions. The following is the relevant extract from the revised manuscript:

*...Moreover, the lack of deviations in the two water towers may indicate the resilience of forested regions (i.e. adaptable nature of forests) as described in (Creed et al., 2014; Helman et al., 2017; Van der Velde et al., 2014). Such resilience (measured as elasticity) could be a key factor in forested water towers indicating their ability to resist change or bouncing back to their initial natural conditions, hence plotting along the reference Budyko curve. Long-term adaptations of forests have been achieved by trees even in the most water-limited forests (Helman et al., 2017). However, our investigations on elasticity (that refers to the degree of initial change using 1981-1990 as the reference period) did not support the above science as lower elasticity values were observed in most of the water towers. Given that low elasticity indicates broad ranges in the evaporative index (EI) compared to the dryness index (DI), this may further indicate the presence of anthropogenic influence within the water towers. According to Creed et al. (2014), elastic catchments are expected to plot along the Budyko curve (i.e. high elasticity = resilient to climate changes) while inelastic catchments (i.e. low elasticity = non-resilience to climate changes) would deviate from the Budyko curve...*

p15 line 10 Isn't this a consequence of the way water towers are defined?

**Answer:** We thank the reviewer for this comment. We have now defined the criteria we applied in our study. The selected water towers followed the criteria where only those

forested areas in high elevated areas and have an Aridity (AI) of 0.65 and above (humid) were selected. We assumed that all the selected water towers would possess humidity conditions given the criteria used. We found this an interesting finding where the two water towers (Mt Meru and Bale mountains) demonstrated warmer conditions (i.e. DI >1). We also discovered climatic shifts between warmer and humid conditions. The following is the relevant extract from the revised manuscript:

*...Further illustrations can be shown in the Budyko space based on the horizontal shifts relative to the dryness index (DI). The horizontal shifts are important indicators of the behavior of the water towers towards warmer or humid conditions. These horizontal deviations reflect a change in the climatic conditions specifically, temperature and precipitation (Creed and Spargo, 2012). This study observed that the majority of the water towers (7 out of 9) plotted within humid conditions (i.e. DI <1). On the other hand, two of the water towers (i.e. Mt Meru and Bale mountains) demonstrated warmer conditions (i.e. DI >1). One major observation is that water towers in Eastern Africa seem to shift towards the left, an indication of the increased humid conditions especially in the period of 2011-2019. At the same time, a gradual increase in PET was observed in all the water towers. A climate shift to wetter conditions and simultaneous increases in regional temperatures have also been reported in the East African region and projected to increase by the end of 21st century (Giannini et al., 2018; Niang et al., 2014; Omambia et al., 2012)...*

*...There are chances that the shifts to wetter conditions in the water towers may also be as a result of the extended impact of increasing PET on the El Nino-Southern Oscillation (ENSO), a phenomenon that influences precipitation in the East African region. Li et al. (2016) investigated annual flood frequencies, from 1990 to 2014, and observed upward trends that were linked to the ENSO phenomenon. Additionally, the shifts to wetter conditions also coincide with the recent reports on the 'rising lake levels phenomenon' in the Eastern Africa region (Chebet, 2020; Chepkoech, 2020; Patel, 2020; Wambua, 2020). We however do not believe we have the data to link the climatic shifts and 'swelling' of lakes to ENSO variations in our study which requires detail scientific investigations...*

**Suggested additional references**

Aleman, J.C., et al. 2018. Forest extent and deforestation in tropical Africa since 1900. Nature ecology & evolution, 2(1), pp.26-33.

El Tom, M.A., 1972. The reliability of rainfall over the Sudan. Geografiska Annaler: Series A, Physical Geography, 54(1), pp.28-31.

Hulme, M., 1990. The changing rainfall resources of Sudan. Transactions of the Institute of British Geographers, pp.21-34.

Ma, X. et al. 2014. Attribution of climate change, vegetation restoration, and engineering measures to the reduction of suspended sediment in the Kejie catchment, southwest China. Hydrology and Earth System Sciences, 18(5), pp.1979-1994.

........................................................................................................................

**References included in the responses**

Aleman, J. C., Jarzyna, M. A. and Staver, A. C.: Forest extent and deforestation in tropical Africa since, Nat. Ecol. Evol., 2(1), 26–33, doi:10.1038/s41559-017-0406-1, 2018.

Astuti, H. P. and Suryatmojo, H.: Water in the forest: Rain-vegetation interaction to estimate canopy interception in a tropical borneo rainforest, IOP Conf. Ser. Earth Environ. Sci., 361(1), doi:10.1088/1755-1315/361/1/012035, 2019.

Bai, P., Zhang, D. and Liu, C.: Estimation of the Budyko model parameter for small basins in China, , 34(1), 125–138, doi:https://doi.org/10.1002/hyp.13577, 2019.

Chebet, C.: Environmental degradation to blame for swelling of Rift Valley lakes, Stand. Media, Kenya [online] Available from: https://www.standardmedia.co.ke/environment/article/2001371606/swelling-lakes-of-the-rift-pose-danger-to-residents, 2020.

Chepkoech, A.: Kenya: Rift Valley Lakes Water Levels Rise Dangerously, Dly. Nation, Kenya [online] Available from: https://allafrica.com/stories/202008310228.html (Accessed 15 May 2021), 2020.

Convention on Biological Diversity: Biodiversity of Dry and Sub-Humid Land Ecosystems, Secr. Conv. Biol. Divers., doi:ISBN: 1020-9387, 2001.

Creed, I. and Spargo, A.: Application of the Budyko curve to explore sustainability of water yields from headwater catchments under changing environmental conditions, in Ecological Society of America, August 5-10, 2012. Portland. [online] Available from: http://www.uwo.ca/biology/faculty/creed/PDFs/presentations/APRE47.pdf, 2012.

Creed, I., Spargo, A., Jones, J., Buttle, J., Adams, M., Beall, F. D., Booth, E. G., Campbell, J. L., Clow, D., Elder, K., Green, M. B., Grimm, N. B., Miniat, C., Ramlal, P., Saha, A., Sebestyen, S., Spittlehouse, D., Sterling, S., Williams, M. W., Winkler, R. and Yao, H.: Changing forest water yields in response to climate warming: Results from long-term experimental watershed sites across North America, Glob. Chang. Biol., 20(10), 3191–3208, doi:10.1111/gcb.12615, 2014.

Dawson, J. B.: The Gregory Rift Valley and Neogene-recent Volcanoes of Northern Tanzania, Geological Society, Memoir 13., 2008.

Dey, P. and Mishra, A.: Separating the impacts of climate change and human activities on streamflow: A review of methodologies and critical assumptions, J. Hydrol., 548, 278–290, doi:10.1016/j.jhydrol.2017.03.014, 2017.

Van Dijk, A. I. J. M., Gash, J. H., Van Gorsel, E., Blanken, P. D., Cescatti, A., Emmel, C., Gielen, B., Harman, I. N., Kiely, G., Merbold, L., Montagnani, L., Moors, E., Sottocornola, M., Varlagin, A., Williams, C. A. and Wohlfahrt, G.: Rainfall interception and the coupled surface water and energy balance, Agric. For. Meteorol., 214–215, 402–415, doi:10.1016/j.agrformet.2015.09.006, 2015.

Donohue, R. J., Roderick, M. L. and McVicar, T. R.: On the importance of including vegetation dynamics in Budyko's hydrological model, Hydrol. Earth Syst. Sci., 11(2), 983–995, doi:10.5194/hess-11-983-2007, 2007.

EAC, UNEP and GRID-Arendal: Sustainable Mountain Development in East Africa in a Changing Climate, East African Community, United Nations Environment Programme and GRID-Arendal. Arusha, Nairobi and Arendal. [online] Available from: https://www.grida.no/publications/119, 2016.

Ellison, D., Morris, C. E., Locatelli, B., Sheil, D., Cohen, J., Murdiyarso, D., Gutierrez, V., Noordwijk, M. van, Creed, I. F., Pokorny, J., Gaveau, D., Spracklen, D. V., Tobella, A. B., Ilstedt, U., Teuling, A. J., Gebrehiwot, S. G., Sands, D. C., Muys, B., Verbist, B., Springgay, E., Sugandi, Y. and Sullivan, C. A.: Trees, forests and water: Cool insights for a hot world, Glob. Environ. Chang., 43, 51–61, doi:10.1016/j.gloenvcha.2017.01.002, 2017.

Gabiri, G., Diekkrüger, B., Näschen, K., Leemhuis, C., van der Linden, R., Mwanjalolo Majaliwa, J. G. and Obando, J. A.: Impact of Climate and Land Use / Land Cover Change on the Water Resources of a Tropical Inland Valley, Climate, 83(8), 1–25, 2020.

Gash, J. H. C., Wright, I. R. and Lloyd, C. R.: Comparative estimates of interception loss from three coniferous forests in Great Britain, J. Hydrol., 48(1–2), 89–105, doi:10.1016/0022-1694(80)90068-2, 1980.

Giannini, A., Lyon, B., Seager, R. and Vigaud, N.: Dynamical and Thermodynamic Elements of Modeled Climate Change at the East African Margin of Convection, Geophys. Res. Lett., 45(2), 992–1000, doi:10.1002/2017GL075486, 2018.

Gunkel, A. and Lange, J.: Water scarcity, data scarcity and the Budyko curve—An application in the Lower Jordan River Basin, J. Hydrol. Reg. Stud., 12(April), 136–149, doi:10.1016/j.ejrh.2017.04.004, 2017.

Han, J., Yang, Y. and Roderick, M. L.: Assessing the Steady - State Assumption in Water Balance Calculation Across Global Catchments Water Resources Research, , (i), 1–16, doi:10.1029/2020WR027392, 2020.

Helman, D., Lensky, I. M., Yakir, D. and Osem, Y.: Forests growing under dry conditions have higher hydrological resilience to drought than do more humid forests, Glob. Chang. Biol., 23(7), 2801–2817, doi:10.1111/gcb.13551, 2017.

Van den Hende, C., Van Schaeybroeck, B., Nyssen, J., Van Vooren, S., Van Ginderachter, M. and Termonia, P.: Analysis of rain-shadows in the Ethiopian Mountains using climatological model data, Clim. Dyn., 56(5–6), 1663–1679, doi:10.1007/s00382-020-05554-2, 2021.

Hillman, J. C.: The Bale Mountains National Park Area , Southeast Ethiopia , and Its Management Author ( s ): Jesse C . Hillman Source : Mountain Research and Development , May - Aug ., 1988 , Vol . 8 , No . 2 / 3 , African Mountains and Highlands ( May - Aug ., 1988 ), , , 8(2), 253–258, 1988.

Hulme, M.: The Changing Rainfall Resources of Sudan, R. Geogr. Soc. ( with Inst. Br. Geogr. ), 15(1), 21–34, doi:https://doi.org/10.2307/623090, 1990.

Hyandye, C. B., Worqul, A., Martz, L. W. and Muzuka, A. N. N.: The impact of future climate and land use/cover change on water resources in the Ndembera watershed and their

mitigation and adaptation strategies, Environ. Syst. Res., 7(1), 7, doi:10.1186/s40068-018-0110-4, 2018.

Jiang, C., Xiong, L., Wang, D., Liu, P., Guo, S. and Xu, C. Y.: Separating the impacts of climate change and human activities on runoff using the Budyko-type equations with time-varying parameters, J. Hydrol., doi:10.1016/j.jhydrol.2014.12.060, 2015.

Keys, P. W., Barnes, E. A., Van Der Ent, R. J. and Gordon, L. J.: Variability of moisture recycling using a precipitationshed framework, Hydrol. Earth Syst. Sci., 18(10), 3937–3950, doi:10.5194/hess-18-3937-2014, 2014.

Kirkby, M., Bracken, L. and Reaney, S.: The influence of land use, soils and topography on the delivery of hillslope runoff to channels in SE Spain, Earth Surf. Process. Landforms, 27(13), 1459–1473, doi:10.1002/esp.441, 2002.

Li, C. juan, Chai, Y. qing, Yang, L. sheng and Li, H. rong: Spatio-temporal distribution of flood disasters and analysis of influencing factors in Africa, Nat. Hazards, 82(1), 721–731, doi:10.1007/s11069-016-2181-8, 2016.

Li, D., Pan, M., Cong, Z., Zhang, L. and Wood, E.: Vegetation control on water and energy balance within the Budyko framework, Water Resour. Res., 49(2), 969–976, doi:10.1002/wrcr.20107, 2013.

Ma, X., Lu, X. X., Van Noordwijk, M., Li, J. T. and Xu, J. C.: Attribution of climate change, vegetation restoration, and engineering measures to the reduction of suspended sediment in the Kejie catchment, southwest China, Hydrol. Earth Syst. Sci., 18(5), 1979–1994, doi:10.5194/hess-18-1979-2014, 2014.

Mango, L. M., Melesse, A. M., McClain, M. E., Gann, D. and Setegn, S. G.: Land use and climate change impacts on the hydrology of the upper Mara River Basin, Kenya: Results of a modeling study to support better resource management, Hydrol. Earth Syst. Sci., 15(7), 2245–2258, doi:10.5194/hess-15-2245-2011, 2011.

Niang, I., Ruppel, O. C., Abdrabo, M. A., Essel, A., Lennard, C., Padgham, J. and Urquhart, P.: Africa. In: Climate Change 2014: Impacts, Adaptation, and Vulnerability, in Part B: Regional Aspects. Contribution of Working Group II to the Fifth Assessment Report of the Intergovernmental Panel on Climate Change. [Barros, V.R., C.B. Field, D.J. Dokken, M.D. Mastrandrea, K.J. Mach, T.E. Bilir, M. Chatterjee, K.L. Ebi, Y.O. Estr, edited by V. R. Barros, C. B. Field, D. J. Dokken, M. D. Mastrandrea, and K. J. Mach, pp. 1199–1265, Cambridge University Press, Cambridge., 2014.

Noordwijk, M. Van, Speelman, E., Hofstede, G. J., Farida, A., Wamucii, C. N., Kimbowa, G., Geraud, G., Assogba, C., Best, L., Tanika, L. and &others: Sustainable Agroforestry Landscape Management:, Land, 9(243), 1–38 [online] Available from: http://dx.doi.org/10.3390/land9080243, 2020.

Odawa, S. and Seo, Y.: Water tower ecosystems under the influence of land cover change and population growth: Focus on mau water tower in Kenya, Sustain., 11(13), doi:10.3390/su11133524, 2019.

Omambia, A. N., Shemsanga, C. and Hernandez, I. A. S.: Climate Change Impacts, Vulnerability, and Adaptation in East Africa (EA) and South America (SA), B. Handb. Clim. Chang. Mitig., 1–4, 573–620, doi:10.1007/978-1-4419-7991-9_17, 2012.

Otieno, V. O. and Anyah, R. O.: Effects of land use changes on climate in the Greater Horn of Africa, Clim. Res., 52(1), 77–95, doi:10.3354/cr01050, 2012.

Patel, K.: Rising Waters on Kenya's Great Rift Valley Lakes, Earth Obs. NASA [online] Available from: https://earthobservatory.nasa.gov/images/147226/rising-waters-on-kenyas-great-rift-valley-lakes (Accessed 15 May 2021), 2020.

Peng, L., Wei, Z., Chen, A., Wood, E. F. and Sheffield, J.: Determinants of the ratio of actual to potential evapotranspiration, , (September 2018), 1326–1343, doi:10.1111/gcb.14577, 2019.

Redhead, J. W., Stratford, C., Sharps, K., Jones, L., Ziv, G., Clarke, D., Oliver, T. H. and Bullock, J. M.: Empirical validation of the InVEST water yield ecosystem service model at a national scale, Sci. Total Environ., 569–570, 1418–1426, doi:10.1016/j.scitotenv.2016.06.227, 2016.

Røhr, P. C. and Killingtveit, Å.: Rainfall distribution on the slopes of Mt Kilimanjaro, Hydrol. Sci. J., 48(1), 65–77, doi:10.1623/hysj.48.1.65.43483, 2003.

Sinha, J., Sharma, A., Khan, M. and Goyal, M. K.: Assessment of the impacts of climatic variability and anthropogenic stress on hydrologic resilience to warming shifts in Peninsular India, Sci. Rep., 8(1), 1–14, doi:10.1038/s41598-018-32091-0, 2018.

Teng, J., Chiew, F. H. S., Vaze, J., Marvanek, S. and Kirono, D. G. C.: Estimation of climate change impact on mean annual runoff across continental Australia using Budyko and Fu equations and hydrological models, J. Hydrometeorol., 13(3), 1094–1106, doi:10.1175/JHM-D-11-097.1, 2012.

Teuling, A. J.: A Forest Evapotranspiration Paradox Investigated Using Lysimeter Data, Vadose Zo. J., 17(1), 170031, doi:10.2136/vzj2017.01.0031, 2018.

Teuling, A. J., De Badts, E. A. G., Jansen, F. A., Fuchs, R., Buitink, J., Van Dijke, A. J. H. and Sterling, S. M.: Climate change, reforestation/afforestation, and urbanization impacts on evapotranspiration and streamflow in Europe, Hydrol. Earth Syst. Sci., 23(9), 3631–3652, doi:10.5194/hess-23-3631-2019, 2019.

El Tom, M. A.: The Reliability of Rainfall over the Sudan, Geogr. Ann. . Ser. A , Phys. Geogr. , 1972 , Vol . 54 , No . 1 ( 1972 ), Publ. by  Taylor Fr. , Ltd behalf Swedish Soc. Anthr, 54(1), 28–31, 1972.

Troch, P. A., Carrillo, G., Sivapalan, M., Wagener, T. and Sawicz, K.: Climate-vegetation-soil interactions and long-term hydrologic partitioning: Signatures of catchment co-evolution, Hydrol. Earth Syst. Sci., 17(6), 2209–2217, doi:10.5194/hess-17-2209-2013, 2013.

UNEP: "Africa Water Atlas". Division of Early Warning and Assessment (DEWA), United Nations Environ. Program. (UNEP). Nairobi, Kenya, 2010.

Van der Velde, Y., Vercauteren, N., Jaramillo, F., Dekker, S. C., Destouni, G. and Lyon, S. W.: Exploring hydroclimatic change disparity via the Budyko framework, Hydrol. Process., 28(13), 4110–4118, doi:10.1002/hyp.9949, 2014.

Wambua, C.: Why Kenya's Rift Valley lakes are going through a crisis, Aljazeera [online] Available from: https://www.aljazeera.com/news/2020/08/30/why-kenyas-rift-valley-lakes-are-going-through-a-crisis/, 2020.

Western, A. W., Zhou, S. L., Grayson, R. B., McMahon, T. A., Blöschl, G. and Wilson, D. J.: Spatial correlation of soil moisture in small catchments and its relationship to dominant spatial hydrological processes, J. Hydrol., 286(1–4), 113–134, doi:10.1016/j.jhydrol.2003.09.014, 2004.

Woods, R.: The relative roles of climate, soil, vegetation and topography in determining seasonal and long-term catchment dynamics, Adv. Water Resour., 30(5), 1061, doi:10.1016/j.advwatres.2006.10.010, 2002.

Yan, D., Lai, Z. and Ji, G.: Using Budyko-type equations for separating the impacts of climate and vegetation change on runoff in the source area of the yellow river, Water (Switzerland), 12(12), 1–15, doi:10.3390/w12123418, 2020.

Yang, H., Qi, J., Xu, X., Yang, D. and Lv, H.: The regional variation in climate elasticity and climate contribution to runoff across China, J. Hydrol., 517, 607–616, doi:10.1016/j.jhydrol.2014.05.062, 2014.

Zhang, M., Wei, X., Sun, P. and Liu, S.: The effect of forest harvesting and climatic variability on runoff in a large watershed: The case study in the Upper Minjiang River of Yangtze River basin, J. Hydrol., 464–465, 1–11, doi:10.1016/j.jhydrol.2012.05.050, 2012.

Zimmermann, L., Frühauf, C. and Bernhofer, C.: The role of interception in the water budget of spruce stands in the Eastern Ore Mountains/Germany, Phys. Chem. Earth, Part B Hydrol. Ocean. Atmos., 24(7), 809–812, doi:10.1016/S1464-1909(99)00085-4, 1999.

---

## Author Comment (AC3)

**RC2**: **'Comment on hess-2021-151'**, Anonymous Referee #2

General comments:

This is a very interesting study; however, I feel that it suffers from two main deficiencies that would need to be addressed in a revised version of the manuscript.

We thank the reviewer for the constructive comments. We have considered all comments and suggestions in our revised manuscript.

1. The first is that there does not appear to be an overarching research question that is being addressed. Given what the authors know about the general climate of eastern Africa, it should be possible to suggest where the various water towers should plot on the baseline Budyko curve, and then test to see if in fact that was the case.

**Answer:** We agree with the reviewer and we have now included a focused research question:

"What are the effects of climate and land-use changes on water yield for the selected forested water towers?"

We tested the following hypothesis:

In areas considered as pristine or protected zones (i.e. high elevated forested areas), with AI≥0.65, changes in water yield would majorly be attributed to climate changes and negligibly due to land use/cover changes. The high elevated forested areas would then be expected to fall on the reference Budyko curve over the study period.

2. The second issue is the absence of any specific consideration of the uncertainties associated with the estimates of the variables (P, PET, NDVI) used in the analysis. My concern here is that the authors spend considerable time discussing temporal changes in water balance components that may or may not fall outside the range of uncertainty associated with these components.

**Answer:** We thank the reviewer for this comment. A discussion on uncertainties has now been included in the discussion section. The following is the relevant extract from the revised manuscript:

*...Besides the strengths in using the Budyko approach, uncertainties may exist which could have affected our results. The study used data from different datasets (CHIRPs, CRU, GIMMS/AVHRR) at various scales which could potentially affect results due to various assumptions and approaches in the processing of each dataset. For instance, the CRU dataset is fairly coarse and contains rather few observations in Africa. One substantial weakness of the current CHIRPS algorithm is the lack of uncertainty information provided by the inverse distance weighting algorithm used to blend the CHIRP data and station data (Funk et al., 2015). The overall NDVI3g uncertainty comes from spatial and temporal coherence variability which gives approximately an error of ±0.002 NDVI units. However, this NDVI error is considered low uncertainty hence applicable to study seasonal and inter-annual non-stationary phenomena (Pinzon and Tucker, 2014). Uncertainties may also arise in the general assumption that estimation of land surface characteristics (ω) based on NDVI formulation provides values that represent integrated conditions for soil, topography, and climate seasonality. Some studies using various hydrological approaches have reported the significance of these factors in influencing catchment hydrology (Kirkby et al., 2002; Troch et al., 2013; Western et al., 2004; Woods, 2002). There is a need for more research to come up with methodological consistency in estimating ω parameters when using the Budyko framework. Although the focus of the study was in the elevated forested areas, empirical adjustment of the Budyko model may be needed to capture special features such as desert wadis in the application of the Budyko equation in the lowland areas.*

*We also recognize other factors that may influence the results in this study. For instance, increasing atmospheric $CO_2$ concentrations may affect terrestrial water cycling through changes in climate and changes in transpiration (i.e. stomatal conductance) (Frank et al., 2015; Huntington, 2008; Mamuye, 2018). We however assume that if $CO_2$ leads to higher NDVI, then this effect is accounted for in our modeling approach. Some studies have reported that NDVI linear trends can be linked to increasing $CO_2$ levels (Krakauer et al., 2017; Yuan et al., 2017). However, detailed investigations are recommended within the East African region. Other factors that may affect our results include the human alteration to water usage. Kiteme et al. (2008) reported unregulated abstraction of water in the upstream of Mt Kenya water tower leading to hydrological droughts downstream. Intensification of irrigated agriculture and a growing human population was reported at the foot slopes of the water towers (Liniger et al., 2005; Ulrich et al., 2012). The effects of anthropogenic presence at the foot slope of the water towers have not been accounted for and further studies are needed to understand how humans living at the footslope of protected water towers affect the pristine conditions of the water towers at high elevations...*

**Specific comments:**

Page/line

3/3                 How was potential evaporation estimated?

**Answer:** Thank you for this comment. We have now included more details on the processing of the datasets used (P, PET, NDVI) in the revised manuscript. The CRU-PET is calculated using the Penman-Monteith formula (Ekström et al., 2007; Harris et al., 2020). The following is the relevant extract from the revised manuscript:

*...Precipitation (P) data were gathered from the Climate Hazards Group Infrared Precipitation with Stations (CHIRPS-v2) with a temporal coverage beginning 1981 and a spatial resolution of 0.05°. CHIRPS uses the Tropical Rainfall Measuring Mission Multi-satellite Precipitation Analysis version 7 (TMPA 3B42 v7) to calibrate global Cold Cloud Duration (CCD) rainfall estimates (Funk et al., 2015). Potential Evapotranspiration (PET) data were sourced from the Climate Research Unit (CRU) database with temporal coverage beginning 1981 and a spatial resolution of 0.5°. The CRU-PET is calculated using the Penman-Monteith formula (Ekström et al., 2007; Harris et al., 2020). Normalized Difference Vegetation Index (NDVI) data to estimate land surface characteristics were sourced from the Global Inventory Monitoring and Modeling System (GIMMS) Third Generation (3g) Advanced Very High-Resolution Radiometer (AVHRR) sensor onboard the National Oceanic and Atmospheric Administration (NOAA) satellites at a spatial resolution of 0.07° (Kalisa et al., 2019; Pinzon and Tucker, 2014; Tucker et al., 2005) The NDVI is derived using the Bayesian methods with high-quality well-calibrated SeaWiFS NDVI data. The resulting NDVI values give an error of ± 0.005 NDVI (Pinzon and Tucker, 2014)...*

3/2-8             What are the uncertainties associated with the estimates of P, PET and NDVI?

**Answer:** We have now included a discussion of the uncertainties and limitations of the study in the revised manuscript.

3/27-30           Can catchments deviate from the Budyko curve under stationary conditions?

**Answer:** Thank you for the comment. Our assumption as mentioned on page 3, line 26 (in the preprint version), is that, under stationary conditions (i.e. naturally occurring fluctuations due to P and PET), catchments will fall on the Budyko Curve, and in nonstationary conditions (i.e. presence of anthropogenic influence), catchments will deviate from Budyko curve. We have revised this part as follows:

*...One important feature of the Budyko curve is the assumption that, under stationary conditions ((i.e. naturally occurring fluctuations due to P and PET), study areas will fall on the Budyko Curve. However, under non-stationary conditions (i.e. anthropogenic influence manifested in ET changes), each catchment will deviate from the Budyko curve depending on land cover and physical catchment characteristics (Creed and Spargo, 2012; Mwangi et al., 2016). This feature can be used to separate land cover change effects from climate change...*

This assumption was critical in testing our hypothesis that in areas considered as pristine or protected zones, changes in water yield would majorly be attributed to climate changes and negligibly due to land use/cover changes. The high elevated forested areas would then be expected to fall on the reference Budyko curve over the study period.

7/Figure 3c     How significant are these changes given the uncertainty associated with w?

**Answer:** We thank the reviewer for this comment. Figure 3C displays the % changes of land surface characteristics using 1981-1990 as the reference period. These changes are significant when connected to the study results especially on the trends of rainfall and water yield over the study period. A discussion on uncertainties associated with $\omega$ has also been included. The following are the relevant extracts from the revised manuscript:

*...The extreme opposite temporal trends observed in water yields from the different water towers confirm a strong variation in the regional climatic patterns. For instance, while there was a consistent increase in annual mean water yield at Mt Elgon, the opposite was true at Mt Kilimanjaro where a steady decline in water yield was observed. Our results further revealed that precipitation (P) is the dominant driver in the East African region. For instance, a consistent increase in Q at Mt Elgon coincided with a steady increase in land surface characteristics ($\omega$) as shown in Figure 3 C. Ideally, a reduction in Q would have occurred due to the increase in ET (associated with increases in land surface characteristics), but this was diffused by the increases in rainfall as shown in Figure 2 C. At Kilimanjaro water tower, a continuous reduction in Q coincided with a steady reduction in $\omega$. Again, an increase in Q would have been expected due to a decrease in ET. Therefore, precipitation is the dominant driver in the generation and supply of blue water from the forested water towers in the East African region...*

*...Uncertainties may also arise in the general assumption that estimation of land surface characteristics ($\omega$) based on NDVI formulation provides values that represent integrated conditions for soil, topography, and climate seasonality. Some studies using various hydrological approaches have reported the significance of these factors in influencing catchment hydrology (Kirkby et al., 2002; Troch et al., 2013; Western et al., 2004; Woods, 2002). There is a need for more research to come up with methodological consistency in estimating $\omega$ parameters when using the Budyko framework. Although the focus of the study was in the elevated forested areas, empirical adjustment of the Budyko model may be needed to capture special features such as desert wadis in the application of the Budyko equation in the lowland areas...*

12/Figure 8     Why is there a general overprediction of Q?

**Answer:** We thank the reviewer for the comment. We have now included a discussion on possible reasons why there is an overestimation of water yield when a comparison with GRDC runoff was done. The following is the relevant extract from the revised manuscript:

*...In this study, the spatial pattern of the simulated streamflow in the Budyko framework closely resembles the pattern observed in the GRDC composite runoff. We however noted overestimation of water yield in the comparison. This type of observation was also reported by (Teng et al., 2012), where the Budyko equation was found to overestimate water yield in drier regions. Moreover, other factors such as soil type, topography, seasonality, water*

*storage, interception, etc were not accounted for in the quantitative framework which can affect the simulations in the selected forested water towers.*

*Canopy interception, for instance, plays an important role in the water balance of forested ecosystems as noted in several studies (Astuti and Suryatmojo, 2019; Gash et al., 1980; Teuling et al., 2019; Zimmermann et al., 1999). In their study, (Teuling et al., 2019) found many forested points to have average yearly evapotranspiration (ET) that exceeds the average potential evapotranspiration (PET). Van Dijk et al. (2015) opined that this is possible due to underestimation of evapotranspiration which was attributed to evaporation of interception water by energy not captured in the formulation of PET. The forest evapotranspiration paradox is further discussed in (Teuling, 2018). The correction of underestimation in (Teuling et al., 2019) indicates the need for long-term lysimeter observations for studies focussing on forested ecosystems. Availability of meteorological data in the upper slopes of the East African mountains is a big gap as the majority of meteorological observations are conducted below 1500 m a.s.l and most of the upper slopes data rely on extrapolation of hydrological analysis in the lowlands (Røhr and Killingtveit, 2003).*

*Local-based runoff measurements would have helped to interpret if there is indeed an overestimation in our study. That said, we observed positive KGE which indicates a "good" model performance (Knoben et al., 2019). Therefore, we considered the Budyko simulations as acceptable. However, it should be noted that this comparison is added for reference only and should not be seen as validation. This is because, the Global composite runoff (Fekete et al., 2002) is not a strictly observational dataset, and it is used here as the "best estimate" available for long-term estimates of streamflow in the East African region...*

12/3-4        How much of the greater sensitivity of water yield to climate changes rather than land use changes is due to the form of equation 1? Is the differential sensitivity simply a function of the formulation of the Budyko curve, or is it real?

**Answer:** We thank the reviewer for this comment. We have extended our discussion by linking our study results to the latest publications. The sensitivity of water yield to climate changes could actually be true and the Buydko equation can therefore be said to simulate the reality. We have also acknowledged the fact that these water towers are under institutional governance hence controlled anthropogenic influence. The following is the relevant extract from the revised manuscript:

*...Our results indicate that changes in precipitation and potential evapotranspiration are the major determinants of blue water availability from high elevated forested water towers in the East African region. Related observations have been made - that climate changes in Africa have a relatively higher impact on water yield compared to other drivers such as land-use changes (Alcamo et al., 2007; Niang et al., 2014). However, it would also be paramount to point out that lack of evidence of sensitivity to land-use changes within the water towers, may also be due to existing institutional arrangements, hence a limited degree of actual change. We presume that the results would be different if such rules would be relaxed...*

15/2-4         "… which further proves the presence of anthropogenic influence …" – does it really "prove" it?

**Answer:** Thank you for the remark. We have replaced the term 'proves' with 'indicates' in the revised manuscript:

19/Figure A4   How can you get different shapes for the baseline Budyko curves for the same w value (e.g. Aberdare Ranges vs. Mt Meru)?

**Answer:** We thank the reviewer for this comment. In our study, each water tower was treated independently in the development of Budyko curves. As explained on page 4 lines 12-20, 100 random points were selected from each of the water towers and assigned

relevant parameters for the calculation of EI and DI indices. Therefore, each water tower has a distinct point-relationship of precipitation (P), potential evaporation (PET), and actual evapotranspiration (ET).

I have attached other specific comments and suggested edits on the manuscript.

**Answer:** Thank you for the supplement with suggested edits. We have revised all areas highlighted.

..................................................................................................................................

**References included in the responses**

Alcamo, J., Flörke, M. and Märker, M.: Future long-term changes in global water resources driven by socio-economic and climatic changes, Hydrol. Sci. J., 52(2), 247–275, doi:10.1623/hysj.52.2.247, 2007.

Astuti, H. P. and Suryatmojo, H.: Water in the forest: Rain-vegetation interaction to estimate canopy interception in a tropical borneo rainforest, IOP Conf. Ser. Earth Environ. Sci., 361(1), doi:10.1088/1755-1315/361/1/012035, 2019.

Creed, I. and Spargo, A.: Budyko guide to exploring sustainability of water yields from catchments under changing environmental conditions, Meet. IAHS-PUB (Prediction Ungauged Basins) Symp. "Completion IAHS Decad. Predict. Ungauged Basins W. ahead," 59, 2012.

Van Dijk, A. I. J. M., Gash, J. H., Van Gorsel, E., Blanken, P. D., Cescatti, A., Emmel, C., Gielen, B., Harman, I. N., Kiely, G., Merbold, L., Montagnani, L., Moors, E., Sottocornola, M., Varlagin, A., Williams, C. A. and Wohlfahrt, G.: Rainfall interception and the coupled surface water and energy balance, Agric. For. Meteorol., 214–215, 402–415, doi:10.1016/j.agrformet.2015.09.006, 2015.

Ekström, M., Jones, P. D., Fowler, H. J., Lenderink, G., Buishand, T. A. and Conway, D.: Regional climate model data used within the SWURVE project projected changes in seasonal patterns and estimation of PET, Hydrol. Earth Syst. Sci., 11(3), 1069–1083, doi:10.5194/hess-11-1069-2007, 2007.

Fekete, B. M., Vörösmarty, C. J. and Grabs, W.: High-resolution fields of global runoff combining observed river discharge and simulated water balances, Global Biogeochem. Cycles, 16(3), 15-1-15–10, doi:10.1029/1999gb001254, 2002.

Frank, D. C., Poulter, B., Saurer, M., Esper, J., Huntingford, C., Helle, G. and Treydte, K.: Water-use efficiency and transpiration across European forests during the Anthropocene, , 5(May), doi:10.1038/NCLIMATE2614, 2015.

Funk, C., Peterson, P., Landsfeld, M., Pedreros, D., Verdin, J., Shukla, S., Husak, G., Rowland, J., Harrison, L., Hoell, A. and Michaelsen, J.: The climate hazards infrared precipitation with stations - A new environmental record for monitoring extremes, Sci. Data, 2, 1–21, doi:10.1038/sdata.2015.66, 2015.

Gash, J. H. C., Wright, I. R. and Lloyd, C. R.: Comparative estimates of interception loss from three coniferous forests in Great Britain, J. Hydrol., 48(1–2), 89–105, doi:10.1016/0022-1694(80)90068-2, 1980.

Harris, I., Osborn, T. J., Jones, P. and Lister, D.: Version 4 of the CRU TS monthly high-resolution gridded multivariate climate dataset, Sci. Data, 7(1), 1–18, doi:10.1038/s41597-020-0453-3, 2020.

Huntington, T. G.: CO2☐induced suppression of transpiration cannot explain increasing runoff, Hydrol. Process., 22(2), 311–314, doi:10.1002/hyp.6925, 2008.

Kalisa, W., Igbawua, T., Henchiri, M., Ali, S., Zhang, S., Bai, Y. and Zhang, J.: Assessment of climate impact on vegetation dynamics over East Africa from 1982 to 2015, Sci. Rep.,

9(1), 1–20, doi:10.1038/s41598-019-53150-0, 2019.

Kirkby, M., Bracken, L. and Reaney, S.: The influence of land use, soils and topography on the delivery of hillslope runoff to channels in SE Spain, Earth Surf. Process. Landforms, 27(13), 1459–1473, doi:10.1002/esp.441, 2002.

Kiteme, B. P., Liniger, H. and Notter, B.: Dimensions of Global Change in African Mountains : The Example of, , 18–22, 2008.

Knoben, W. J. M., Freer, J. E. and Woods, R. A.: Technical note: Inherent benchmark or not? Comparing Nash-Sutcliffe and Kling-Gupta efficiency scores, Hydrol. Earth Syst. Sci., 23(10), 4323–4331, doi:10.5194/hess-23-4323-2019, 2019.

Krakauer, N. Y., Lakhankar, T. and Anadón, J. D.: Mapping and Attributing Normalized Difference Vegetation Index Trends for Nepal, , 1–15, doi:10.3390/rs9100986, 2017.

Liniger, H., Gikonyo, J., Kiteme, B. and Wiesmann, U.: Assessing and Managing Scarce Tropical Mountain Water Resources, Mt. Res. Dev., 25(2), 163–173, doi:https://doi.org/10.1659/0276-4741(2005)025[0163:AAMSTM]2.0.CO;2., 2005.

Mamuye, M.: Review on Impacts of Climate Change on Watershed Hydrology, , 8(1), 91–99, 2018.

Mwangi, H. M., Julich, S., Patil, S. D., McDonald, M. A. and Feger, K. H.: Relative contribution of land use change and climate variability on discharge of upper Mara River, Kenya, J. Hydrol. Reg. Stud., 5, 244–260, doi:10.1016/j.ejrh.2015.12.059, 2016.

Niang, I., Ruppel, O. C., Abdrabo, M. A., Essel, A., Lennard, C., Padgham, J. and Urquhart, P.: Africa. In: Climate Change 2014: Impacts, Adaptation, and Vulnerability, in Part B: Regional Aspects. Contribution of Working Group II to the Fifth Assessment Report of the Intergovernmental Panel on Climate Change. [Barros, V.R., C.B. Field, D.J. Dokken, M.D. Mastrandrea, K.J. Mach, T.E. Bilir, M. Chatterjee, K.L. Ebi, Y.O. Estr, edited by V. R. Barros, C. B. Field, D. J. Dokken, M. D. Mastrandrea, and K. J. Mach, pp. 1199–1265, Cambridge University Press, Cambridge., 2014.

Pinzon, J. E. and Tucker, C. J.: A non-stationary 1981-2012 AVHRR NDVI3g time series, Remote Sens., 6(8), 6929–6960, doi:10.3390/rs6086929, 2014.

Røhr, P. C. and Killingtveit, Å.: Rainfall distribution on the slopes of Mt Kilimanjaro, Hydrol. Sci. J., 48(1), 65–77, doi:10.1623/hysj.48.1.65.43483, 2003.

Teng, J., Chiew, F. H. S., Vaze, J., Marvanek, S. and Kirono, D. G. C.: Estimation of climate change impact on mean annual runoff across continental Australia using Budyko and Fu equations and hydrological models, J. Hydrometeorol., 13(3), 1094–1106, doi:10.1175/JHM-D-11-097.1, 2012.

Teuling, A. J.: A Forest Evapotranspiration Paradox Investigated Using Lysimeter Data, Vadose Zo. J., 17(1), 170031, doi:10.2136/vzj2017.01.0031, 2018.

Teuling, A. J., De Badts, E. A. G., Jansen, F. A., Fuchs, R., Buitink, J., Van Dijke, A. J. H. and Sterling, S. M.: Climate change, reforestation/afforestation, and urbanization impacts on evapotranspiration and streamflow in Europe, Hydrol. Earth Syst. Sci., 23(9), 3631–3652, doi:10.5194/hess-23-3631-2019, 2019.

Troch, P. A., Carrillo, G., Sivapalan, M., Wagener, T. and Sawicz, K.: Climate-vegetation-soil interactions and long-term hydrologic partitioning: Signatures of catchment co-evolution, Hydrol. Earth Syst. Sci., 17(6), 2209–2217, doi:10.5194/hess-17-2209-2013, 2013.

Tucker, C. J., Pinzon, J. E., Brown, M. E., Slayback, A., Pak, E. W., Mahoney, R., Vermote, E. F. and Saleous, N. E. L.: An extended AVHRR 8-kni NDVI dataset compatible with MODISand SPOT vegetation NDVI data, Int. J. Remote Sens., 26(20), 2005.

Ulrich, A., Ifejika Speranza, C., Roden, P., Kiteme, B., Wiesmann, U. and Nüsser, M.: Small-scale farming in semi-arid areas: Livelihood dynamics between 1997 and 2010 in Laikipia, Kenya, J. Rural Stud., doi:10.1016/j.jrurstud.2012.02.003, 2012.

Western, A. W., Zhou, S. L., Grayson, R. B., McMahon, T. A., Blöschl, G. and Wilson, D. J.: Spatial correlation of soil moisture in small catchments and its relationship to dominant spatial hydrological processes, J. Hydrol., 286(1–4), 113–134, doi:10.1016/j.jhydrol.2003.09.014, 2004.

Woods, R.: The relative roles of climate, soil, vegetation and topography in determining seasonal and long-term catchment dynamics, Adv. Water Resour., 30(5), 1061, doi:10.1016/j.advwatres.2006.10.010, 2002.

Yuan, W., Piao, S., Qin, D., Dong, W., Xia, J., Lin, H. and Chen, M.: Influence of Vegetation Growth on the Enhanced Seasonality of Atmospheric $CO_2$, Global Biogeochem. Cycles, 32, 32–41, doi:https://doi.org/10.1002/ 2017GB005802, 2017.

Zimmermann, L., Frühauf, C. and Bernhofer, C.: The role of interception in the water budget of spruce stands in the Eastern Ore Mountains/Germany, Phys. Chem. Earth, Part B Hydrol. Ocean. Atmos., 24(7), 809–812, doi:10.1016/S1464-1909(99)00085-4, 1999.